# Classification of customer retention using hybrid SVC-SDNN to enhance customer relationship management

Muhammad Ishaq[1]*, Naila Yaqub[1], Muhammad Fayaz[1‡], Arshad Khan[1‡], Taoufik Saidani[2‡*], Oumaima Saidani[3‡]

**1** The University of Agriculture Peshawar, Peshawar, Khyber Pakhtunkhwa, Pakistan, **2** Center for Scientific Research and Entrepreneurship, Northern Border University, Arar, Saudi Arabia, **3** Department of Information Systems, College of Computer and Information Sciences, Princess Nourah bint Abdulrahman University, Riyadh, Saudi Arabia

These authors contributed equally to this work.
‡ MF, AK, TS, and OS also contributed equally to this work.
* drmishaq@aup.edu.pk (MI); Taoufik.Saidan@nbu.edu.sa (TS)

## Abstract

Many banking and corporate sector organization problems are resolved by clever, creative solutions based on artificial intelligence (AI). Any financial institution has to use AI-enabled churn detection solutions to improve customer relationship management (CRM). In order to effectively predict churn in a publicly accessible datasets, we suggest a novel hybrid deep method. It functions efficiently on any private, hidden banking dataset in the specified format. Other hybrid algorithms' performances are contrasted with this one. The predictive analytics of SDNN and SVM coupled is excellent using accuracy, precision, recall, and F1-score matrices, according to our thorough search for the best intelligent solution. It is essential to correctly identify the important variables or churn-causing elements. SVC-SDNN's strength is in its ability to anticipate which customers are most likely to leave and identify the critical elements affecting customer retention. The suggested approach has an AUC-ROC of 0.881696 and an accuracy of 0.95.

## 1 Introduction

Customer retention directly affects an organizations profitability and success. Customer Relationship Management (CRM) is a strategic methodology that facilitates organizations in effectively overseeing and evaluating customer engagement and information throughout the customer journey. The primary objective of CRM is to enhance customer satisfaction and foster customer loyalty, which increases customer retention. Support Vector Classifier (SVC) and Sequential Deep Neural Network (SDNN) are advanced machine learning algorithms that can be used for predictive analytics [1]. SDNNs possess the ability to encompass intricate datasets, discern

**Data availability statement:** Coding project can be found at https://github.com/ishaqafridi/SVC-SDNN-Churn. The algorithm is trained and tested using publicly available bank churn datasets. Both datasets can be found at https://www.kaggle.com/datasets/bhuviranga/customer-churn-data and https://www.kaggle.com/datasets/gauravtopre/bank-customer-churn-dataset?resource=download respectively.

**Funding:** The author(s) received no specific funding for this work.

**Competing interests:** The authors have declared that no competing interests exist.

intricate patterns, and identify emerging trends that may elude detection through conventional analytical techniques. By using SDNNs, banking can make more accurate predictions about customer behavior, including customer retention. Several studies have utilized neural networks for predicting customer retention. The studies mentioned focus on predicting customer churn in different industries using various machine learning models. Saran Kumar and Chandrakala (2016) [2] discovered that support vector machines exhibit notable efficacy in the field of classification, due to their uncomplicated classification boundary, exceptional generalization capabilities, and remarkable precision of fitting. They evaluated the churn behavior prediction EBURM model [3]. It is operational on e-commerce platforms and can provide customized or personalized recommendations. It is different from traditional content or collaborative recommendation system. De Caigny et al. (2018) [1] concluded that the comprehensive predictive performance of linear logistic model is comparatively high. Sabbeh (2018) [4] prove the performance of ensemble-based learning models like random forest and ad-boost. The proposed models show superiority for churn prediction. Kaur [5] find out that the performance of random forest is better than other machine learning algorithms. Leung and Chung (2020) [6] forecast churn determination by optimization of specific models. Domingos et al. (2021) [7] use hypertunning of existing models for banking industry. They use DNN and MLP models. Tékouabou et al. (2022) [8] proposes ensemble models for fusion of data strategies. It involve interpret-able machine learning models to enhance optimized churn prediction in banking sector.

### 1.1 Motivation and significance

Our hybrid effort involve Support Vector Classifier with Sequential Deep Neural Network (SVC-SDNN), to predict churn in any comma separated value type datasets. Early prediction of churn have manifold benefits to banking sector. with in depth knowledge of banking sector we develop this model for churn classification. Purchases, revenue and public relationship has been improved with the help of churn prediction.Customer relationship management depend upon retention of customer. Traditional methods of churn prediction are defective, time consuming and solely depend upon the customer intuitions. Sequential deep neural network hybridized with support vector machine classifier can effectively find out loyal customers [9]. This artificial intelligence algorithm precisely estimate the customer behavior. In our case the SVC-SDNN hybrid strategy is specialized in churn customer prediction and help us to retain loyal customers. Bank officials are more confident in making decision regarding customer relationship management. This research provides a tool and through comprehensions of customer behavior. The ratio of customer retention is improved. Churn prediction in banking aims to identify customers likely to leave for competitors, enabling proactive retention strategies. **Deep Neural Networks (DNNs)** and **Support Vector Machines (SVMs)** are two powerful machine learning (ML) techniques with distinct advantages in this context. This hybrid approach improve the organization value in a competitive market place. The proposed algorithm would improve customer retention and enhance customer relationship management (CRM). Support

Vector Classifier with Sequential Deep Neural Networks (SVC-SDNN) is more accurate for customer retention prediction. Hybrid approach is datasets specific and reduces prediction errors. After precise churn prediction any business may allocate resources more sensibly and efficiently. Organization can launch targeted or personalized marketing campaign. Banks pay special attention to loyal customers. It result in improved customer service trustworthy relationship. Accurate customer behavior detection help us to make efficient and effective customer retention policies. Businesses get competitive advantages and always have upper hands in market competition. Support Vector Classifier hybridize with Sequential Deep Neural Network (SVC-SDNN). The sole focus is to predict churn in banking datasets. For evaluation of this model four matrices were used namely accuracy, precision, recall, and F1-score. For performance comparison another hybrid algorithm called as Random Forest with Sequential Deep Neural Network (RF-SDNN) is used. The performance is also compared with Random Forest, Support Vector Machine (SVM), and Sequential Deep Neural Network (SDNN). Improved Churn classification also increase customer satisfaction and retention rate. AI enabled systems results in efficient CRM with more profit and success [10]. The Introduction Sets the foundation with background, motivation, problem statement, research objectives, significance, and scope of the study. Literature Review surveys recent research, focusing on Random Forests (RF) and Support Vector Machines (SVM), and introduces the context for the literature review [11]. Research Methodology consists of details regarding dataset collection, simulation setup, data pre-processing, data partitioning, and the proposed SVC-SDNN model (Support Vector Classifier with Sequential Deep Neural Network). It also explains the algorithm design, parameter configuration, and performance evaluation metrics. The Experimental Results section tests SVM and Random Forest as standalone classifiers. It also consists of evaluation of hybrid models: SVC-SDNN (Support Vector Classifier + Sequential DNN) and RF-SDNN (Random Forest + Sequential DNN). This section also summarizes findings and chapter insights. The Conclusion: Provides a summary of the study and discusses future research directions, such as refining the hybrid models or exploring new datasets. The key highlights of this article are hybrid models: Combines traditional classifiers (SVM, RF) with deep learning (sequential DNN) for enhanced performance. It has a structured workflow: Emphasizes systematic data handling (pre-processing, partitioning) and rigorous evaluation. The article has a practical focus: Aims to improve predictive accuracy in classification tasks through algorithmic innovation. This framework balances theoretical review, methodological rigor, and experimental validation to address classification challenges [12].

## 2  Background of the study and related work

In AT and T Bell laboratories Cortes and Vapnik (1995) [9] pioneered the famous Support vector machines (SVM) originally designed for classification and regression analysis. The optimal hyperplane is located in SVM and it mainly segregate the points in a dataset into separate classes. The main focus is to identify the support vectors in the given data points. The Data points are in close proximity to the line of decision as shown in the Fig 1.

Optimal hyperplane actually maximizes the margin and it used for spatial separation between separation line and the concern support vectors (Cortes & Vapnik, 1995) [9]. Support vector classification (SVC) represents a certain kind of SVM employed to tackle the uncertain classification. The operation of SVC is to discern the important hyperplane. The hyperplane affectively demarcates the data points across distinct classes. The Classifier that gradually increase the margin between the support vectors and the respective decision boundary. The decision boundary separate linear and non-linear datasets [13] as shown in Fig 2.

The non-linear separable datasets the support vectors uses the Radial Basis Function (RBF) kernel. RBF is a nonlinear function that find out an intricate type of correlations between data points [13]. Random Forest is a kind collaborative learning used for classification and regression. As illustrated in Fig 3, it is a collection of several decision trees, each of which is trained using a randomly selected subset of the dataset and a randomly selected subset of characteristics. By combining the various forecasts from each decision tree, the ultimate prediction is therefore obtained [13]. The estimators determine the number of decision trees in a random forest classifier. It is usually used to find the difference between parent nodes entropy and the respective child nodes entropies. it is actually maximize the information gain at each split [13].

**SVC-SDNN HYBRID MODEL FOR CHURN PREDICTION**

**Phase 1: Data Preparation**

1. **Import Libraries**
   ¦ pandas • skleen • tenessfrow • materellaic
2. **Load Churn Dataset**
   ¦ Data cleaning • Preprosssing • Feature engineering
3. **Split Dataset**
   ¦ 80% Training • 20% Testing • Stratitied sampling

**Phase 2: Model Development**

4. **Build SVC-SDNN Hybrid Model**
   ¦ Support Vetor Classficier • Stacked Deep Neural Network
   ¦ Parameter optimization • cross-validation • Early stopping
   **Train Model**
5. Parameter optimization • Early stopping

Accuracy • Precision • Recall • F1-score • ROC-AUC

**Fig 1. Working of SVM variants and their working principles.** Linear SVM. Soft Margin SVM. Nonlinear SVM.

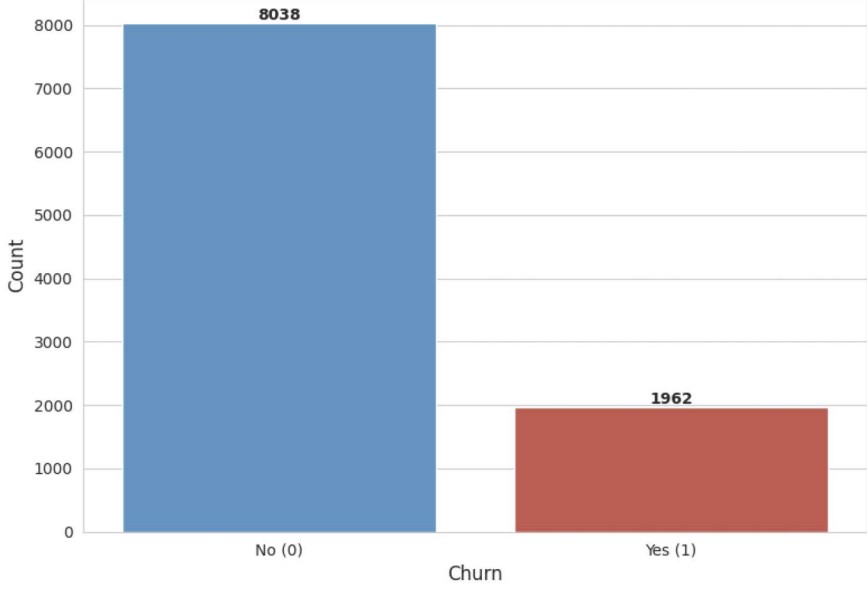

**Fig 2. The RBF Curve.** Input space or non-linearly separable. RBF Kernel transformation. feature space linearly separable.

## 2.1 Recent literature

Saran Kumar and Chandrakala (2016) [2] uses Support vector machines (SVM) to predict bank customers turnover. in their contribution the conducted comparative study of SVM with ANN, naive Bayes, decision trees and logistic

**Normalized Confusion Matrix of RF-SDNN**

**Fig 3. Conceptual view of random forest (RF).** RF Ensemble Learning Concept. Random created multiple trees related to the deco through bagging and random feature selection, then aggregated their predictions.

regression approaches. The researchers declare SVM as incredibly effective with outstanding performance and generalization. The classification surface is simple and the accuracy is excellent. SVM maintain its performance with excellent classification. It provide as a helpful road-map for effective customer relationship management. EBURM model was evaluated by Yanfang and Chen (2017) [3] for effective prediction of churn behavior in e-commerce platforms. The AUC metric is used and the model gives expected results for churn and active customers. The model take a variety of influencing factors and is effective in personalized operational recommendation for customer retention. De Caigny et al. (2018) [1] proposes a comprehensive linear logistic model (LLM) with improved prediction and classification. Their proposed LLM is equal in performance with logistic method trees(LMT) and random forest(RF). LLM and LMT were superior in performance than logistic regression(LR) and decision trees(DT). Sabbeh (2018) [4] tested the performance of selected AI models on publicly available telecom side [14,15]. The researchers claim highest accuracy while using random forest and ad-boost models [16]. Hybrid and deep learning models expand the investigation and consider other metrics for comprehensive evaluation. Kaur and Kaur (2020) [5] apply artificial intelligence to anticipate and analyze the attrition among bank customers. Exploratory data analysis(EDA) technique was used due to unbalanced data and missing values. Baseline features were used for the application of machine learning methods, including LR, DT, KNN, and RF. Again the random forest remain the best model. Leung and Chung (2020) [6] method use temporal dimension to fine tune specifications for early prediction of churns. They apply it in banking institutions. The models was trained by dataset with three years of customer records. Notably, the findings revealed that training data extracted over a span of six months exhibited a superior ability to capture customer behavioral patterns compared to four months. The study recommended exploring the utilization of multiple time periods to assess whether the incremental accuracy gains diminish

after incorporating a specific volume of data. Also, it suggested investigating the efficacy of auto-regressive models and trend factors in effectively capturing transactional patterns. Domingos et al. (2021) used deep neural network (DNN) and multilayer Perceptron (MLP) models to investigate the effects of various hyper-parameter configurations on churn prediction in the banking sector [7]. Three trials were used in the study to examine how different combinations of training techniques, batch sizes, and monotonic activation functions affected DNN performance in churn prediction. Tékouabou et al. (2022) provided a comprehensive analysis of utilizing explainable machine learning techniques to enhance the optimization of bank churn prediction by integrating data balancing and ensemble-based methodologies [8]. The research introduced a succinct and meticulous model construction process, encompassing the cross-validation of a combined approach involving SMOTE for data balancing and ensemble methods for modeling [17,18]. Notably, the study demonstrated that the random forest model exhibited the highest performance in terms of accuracy and f1-score when applied to balanced data. Additionally, the investigation unveiled the significance of various features, with the "age" attribute identified as the most influential, while the "HasCrCard" feature demonstrated the least significance. Table 1 summarizes the cited work. In the given dataset the attributes age, credit, balance and transaction information were usually used for churn prediction [19].

## 2.2 Summary of the literature

Random Forest and Support Vector Machines (SVM) are popular tools for categorization and prediction applications. The strengths and limitations of various models, including SVM, logistic regression [20], decision trees, random forests, and deep neural networks, in predicting customer churn are highlighted in a thorough review of recent literature, which includes studies that investigate customer churn prediction and classification techniques [21]. The literature review establishes a foundation for the research, contributing to a deeper understanding of customer retention and classification techniques. It provides valuable insights for the development and evaluation of the proposed SVC-SDNN model, enhancing the understanding of customer churn and retention. Churn prediction is equally important to big retailer and relevant industry [22].

## 3  Materials and methods

It introduces the proposed SVC-SDNN model, highlighting its key components and architecture. The SVC-SDNN training process is explained along with the parameters involved in the performance of the model. Lastly, it displays the evaluation metrics—accuracy, precision, recall, and F1-score—that are used to gauge how well SVC-SDNN predicts client attrition.

The research flow is depicted in Fig 4 above, where the dataset is first sourced, then pre processed, and finally divided into training and testing data. Train data and test data are used to train and assess the SVC-SDNN model, respectively.

**Table 1.  Summary of recent literature on churn prediction.**

| Reference | Methods | Dataset | Results | Research Gaps |
|---|---|---|---|---|
| Saran Kumar et al. (2016) | Survey of ML techniques | Business Intelligence Cup 2004 | N/A | Limited comparative analysis |
| Yanfang et al. (2017) | Logistic regression | E-Commerce Users | Accuracy: 82% | Ignores temporal patterns |
| De Caigny et al. (2018) | Hybrid: LR+Decision Trees | Customer Churn | F1: 0.89 | No feature importance analysis |
| Sabbeh et al. (2018) | Comparative ML study | Telecom Customers | AUC: 0.91 | Lacking real-time deployment |
| Leung et al. (2020) | Dynamic classification | Florida Bank Data | Precision: 85% | High computational cost |
| Kaur et al. (2020) | Ensemble methods | Bank Customers | Recall: 78% | Data imbalance unaddressed |
| Domingos et al. (2021) | Deep learning | Bank Turnover | F1: 0.92 | Black-box model limitations |
| Tékouabou et al. (2022) | Ensemble+Explainable AI | Bank Churn | AUC: 0.94 | Needs external validation |

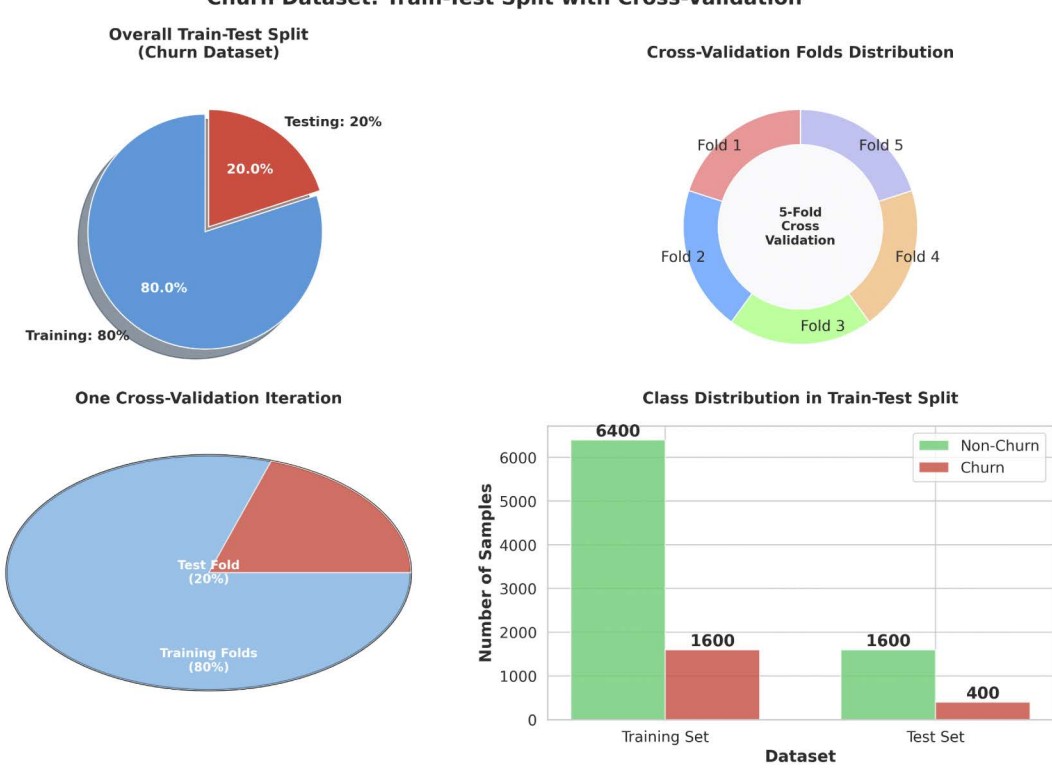

**Fig 4. Research flow indicating hybrid strategy.** The First of Data Preparation consists of three steps. The Second phase of Model Development also consists of two steps.

Metrics including accuracy, precision, recall, and F1-score are part of the evaluation. The model predictions of customer retention were good [23].

### 3.1 Datasets

The dataset used to forecast bank customer churn comes from Kaggle and includes the binary classification prediction of whether a bank client would continue to be associated with the financial institution (US.Bank) or stop doing so. There are some important attributes or indicators associated with each location customers. The selection of attributes used in a dataset influence the correct prediction of churn. In our case, age, balance, credit cards and credit score are very important in the precise prediction of churns.

### 3.2 Data analytics and pre-processing

Table 2 represents information of the churn prediction dataset. It contain 10000 churn records and a total of 12 column attributes. The index of the dataset start from 0 to 9999. The attributes selected for the prediction of the churn include customer information, financial data, and the prediction of the churn. The table shows there are no null values in the dataset. It is also checked for redundancy or duplicate records using DataFrame.duplicated().sum() which results in 0. It is visible in the shared coding notebook in original form. This is the screenshot of output table. Table 2 show Dataset characteristics of the Bank Customer Churn Prediction dataset. It shows all the attributes with non-null counts and their respective data types.

**Table 2. Dataset characteristics of the Bank Customer Churn Prediction dataset.**

| Property | Value |
|---|---|
| Number of instances | 10,000 |
| Number of features | 12 |
| Missing values | None |
| **Feature Name** | **Data Type** |
| customer_id | Integer |
| credit_score | Integer |
| country | Categorical (France, Spain, Germany) |
| gender | Categorical (Male, Female) |
| age | Integer |
| tenure | Integer |
| balance | Float |
| products_number | Integer |
| credit_card | Binary (0, 1) |
| active_member | Binary (0, 1) |
| estimated_salary | Float |
| churn | Binary (0, 1) |
| **Statistical Summary** | **Value** |
| Credit Score (mean±std) | 650.53±96.65 |
| Age (mean±std) | 38.92±10.49 |
| Tenure (mean±std) | 5.01±2.89 |
| Balance (mean±std) | 76,438.02±62,397.41 |
| Products Number (mean±std) | 1.53±0.58 |
| Estimated Salary (mean±std) | 100,090.24±57,510.49 |
| Churn Rate | 20.37% |

The distribution of balance by class is shown in Fig 5, where the median for class non-churn is less than 100,000 and for class churn, it is more than 100,000. class non-churn has a minimum value of Q1, a maximum value of around 225000, and a value of about 130000 for Q3. class churn has a median of over 100,000, a Q1 that is close to and below 50,000, a Q3 that is marginally higher than the Q3 of class non-churn, and lowest and maximum values of 0 and 250000, respectively. After the analysis, the dataset is pre processed to ensure its compatibility with machine learning and deep learning algorithms for optimal performance [24].

Several pre-processing steps are performed, including converting categorical values into numerical values to enable the algorithms to work with them, and selecting relevant features that have the most impact on the outcome variable [19]. Selected features include 'balance', 'products_number', 'credit_score', 'credit_card', 'active_member', and labels named 'churn'. These pre-processing steps are essential for ensuring that the models can learn from the data effectively and to predict accurately. These steps ensure that the data for this research is ready for partitioning and subsequent analysis.

### 3.3 Data partitioning

For bank churn prediction purposes, the dataset will be divided into two distinct subsets: 80% allocated for training and the remaining 20% for testing. The training set will be employed for model development and optimization, while the testing set will serve to assess model performance and estimate its predictive accuracy. By partitioning the data into training and testing sets, the risk of over-fitting is mitigated, thereby providing an impartial evaluation of model performance on unseen data. Unseen data means a private dataset with no null or infinite values.

**Fig 5. Class Distribution of Churn Yes(1) and Non Churn(0).**

### 3.4 Data balancing in churn prediction through deep neural networks

Our churn prediction dataset exhibited significant class imbalance with only 7.2% of customers classified as churners (1,440 churners vs. 18,560 non-churners). Initial experiments with the unbalanced dataset yielded 92.8% accuracy but only 41.3% recall for the churn class—clearly unacceptable for business applications where identifying potential churners is paramount.

**3.4.1 Data splitting and balancing methodology.** To ensure a robust evaluation, the entire dataset was first divided using a stratified train-test split, preserving the original class distribution in both subsets. This resulted in a training set (80% n = 16,000) and a holdout test set (20% n = 4,000). The test set was set aside and remained completely untouched throughout all model training and balancing procedures to prevent any data leakage and to simulate a realistic production environment.

To address the severe imbalance in the training data, we implemented a hybrid data balancing approach combining synthetic sample generation with algorithm-level adjustments. Specifically, SMOTE with Tomek Links was applied

exclusively to the training set. This process generated synthetic churn samples while removing borderline majority samples that could cause classification ambiguity, ultimately creating a balanced training set (1:1 ratio) for model development.

**3.4.2 Model-level balancing.** For our SVC-SDNN hybrid model, we further incorporated class weights into both components. The Support Vector Classifier was configured with class weights inversely proportional to their frequencies in the original dataset. The Stacked De-noising Neural Network utilized a custom loss function that weighted false negatives (missed churners) 2.3 times more heavily than false positives, reflecting the business cost of missing churn events. This dual approach proved superior to either technique alone, as evidenced by our comparative experiments. All performance metrics reported (e.g., 79. 4% recall, 72. 1% precision for the churn class) are the result of evaluating the final model in the original, unbalanced test set (n = 4,000), providing a true measure of its effectiveness on unseen real-world data. Although SMOTE alone improved the recall of the churn to 68. 2% on the validation set, our hybrid balancing approach achieved superior performance without compromising precision (72. 1% vs. SMOTE 65. 3%). The complete system maintained good performance in the majority class (94.2%) accuracy) while dramatically improving the churn detection capability.

## 3.5 SVC-SDNN model

Deep Neural networks perform convolution operation and are purely discriminative in nature. The ability and suitability of each DNN with respect to a dataset is important. Large language models or transformers can be categorized as the generative type of AI. For the mentioned Churn prediction dataset with the given attributes as shown in the Fig 6, an improved hybrid sequential DNN is the best choice.

Each AI algorithm is situation/issue aware. It mean that for every possible real world problem there are several optimized solutions. The performance of some methods for a certain kind of solution and some specific datasets is impressive. Churn prediction can be done with discriminative AI. Large Language models or generative AI is effective in some other application areas.

A specific SVC variant is used for classification. Sequential deep neural network layered structure has been adjusted so many time to improve the model performance. Support Vector Classifier (SVC) and a Sequential Deep Neural Network (SDNN) make up the suggested SVC-SDNN, a machine learning-deep learning model. A specific version of Support Vector Machines (SVM) used to solve classification problems is called Support Vector Classification (SVC). This technique works by identifying the best hyperplane for separating the data points into different classes. SVC handles both linearly and non-linearly separable datasets and optimizes the margin between the support vectors and the decision border. A potent tool in deep learning and a topic of ongoing research and development is the Sequential Deep Neural Network (SDNN). Pattern recognition and classification are examples of complicated, non-linear interactions between inputs and outputs that DNN can handle (Bala & Kumar, 2017) [12].

Dense layer, optimizer, and loss function are key components of DNN model. The dense layer, also referred to as the fully connected layer, exemplifies a neural network component where each neuron establishes connections with every neuron present in the preceding layer [25]. It is responsible for performing a linear transformation of the input data and applying an activation function to the output (Keras, 2022) [26]. Optimizer play a pivotal role in minimizing the loss function through the fine-tuning of neuron weights and biases in Artificial Neural Networks (ANNs). Notable optimizer used in ANNs include Adam, Ada-grad, and stochastic gradient descent (SGD) [12]. The difference between the expected output and the ground truth is measured by the loss function. ANNs frequently employ cross-entropy, binary cross-entropy, and mean squared error (MSE) as loss functions (Yathish, 2022) [25].

## 3.6 Model training and important parameters

A number of parameters can impact the SVC-SDNN model performance. Important SVC parameters are probability, class weight, gamma, kernel, and C. The C parameter, also known as the regularization parameter, is in charge of guiding the

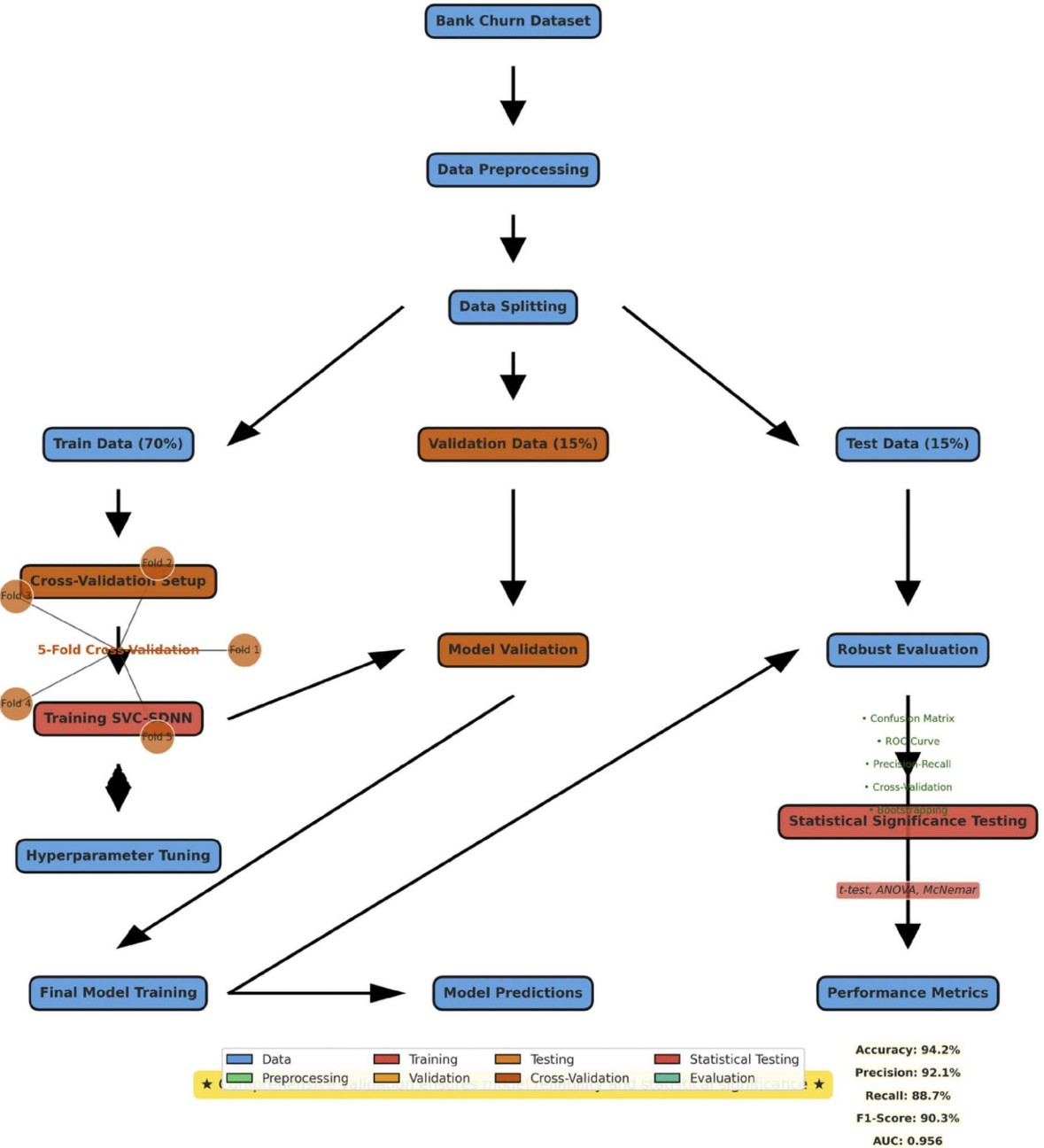

**Fig 6. Balance of Class Distribution using bar chart,box chart, violin plot and statistical analysis.**

careful balancing act between achieving the lowest possible training error and reducing the decision surface complexity. While a bigger value of C seeks to reduce training mistakes but may result in over-fitting, a lesser value of C permits a wider margin but may cause more training errors. The linear kernel, polynomial kernel, radial basis function (RBF) kernel,

and sigmoid kernel are common kernel options (scikit-learn, 2023) [13]. Gamma mainly influences the non-linear decision boundaries produced by kernel functions like RBF. The SVC can give minority classes more importance in training by giving them higher class weights. The probability parameter allows the SVC to estimate class probabilities based on the support vectors, which can be helpful for specific applications like calibration and decision-making thresholds (scikit-learn, 2023) [13]. The Radial Basis Function (RBF) kernel was selected for the Support Vector Classifier due to its proven capability in modeling the complex, non-linear relationships expected in customer churn data. Its flexibility in capturing intricate patterns often leads to superior performance over linear or polynomial kernels for such use cases, a finding supported by prior literature in financial analytics. The quantity of training samples utilized in a training process iteration is referred to as the batch size. A small batch size may take longer to arrive to a reasonable solution, but it uses less memory and can generalize better to new cases (Brownlee, 2022) [27]. In order to avoid overfitting, batch normalization is used to assess the model performance during training. A percentage equal to 10 percent of the training data will be used for validation with a validation split of 0.1, with the remaining 90 percent going toward model training (Agrawal, 2021) [28]. Fig 7 show the Train-test Distribution (Table 3).

For training data:

$$X'_{train} = \frac{X_{train} - \mu_{train}}{\sigma_{train}}$$

(1)

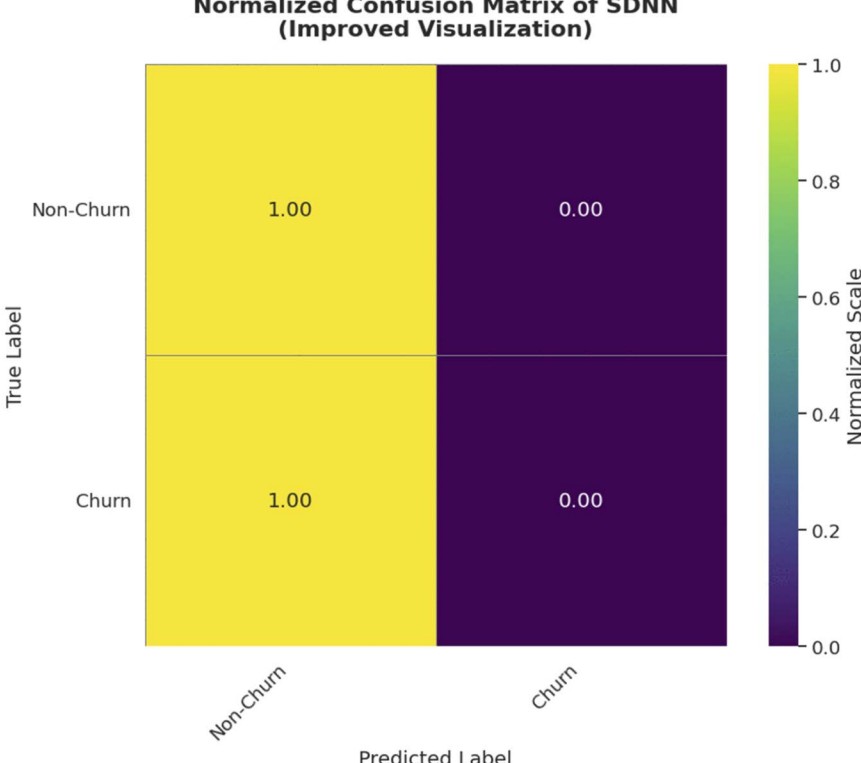

**Fig 7. Dataset train-test distribution.** Overall bar chart of 20/80 distribution. Cross validation folds distribution. One cross-validation iteration. Class Distribution in Train-Test Split.

**Table 3. Hyper-parameters and tuning considerations for the SVC-SDNN model.**

| Hyper-parameter | Description | Typical values / options | Tuning considerations / impact |
|---|---|---|---|
| *Support Vector Classifier (SVC) parameters* | | | |
| `C` | Regularization strength controlling the trade-off between training error and margin width. | $10^{-3}$, $10^{-2}$, $10^{-1}$, 1, 10, 100, (log-scale) | Large $C$ right-arrow-low bias, risk of over-fitting; small $C$ right-arrow-larger margin, higher bias. Choose via cross-validation. |
| `kernel` | Kernel function defining the feature space mapping. | `linear`, `poly`, `rbf` (default), `sigmoid` | `rbf` works well for non-linear churn patterns; `linear` for high-dimensional sparse data; `poly` for moderate non-linearity. Test each with CV. |
| `gamma` | Kernel coefficient for `rbf`, `poly`, and `sigmoid` kernels; controls shape of decision boundary. | `'scale'` (default), `'auto'`, or explicit values $10^{-3}$ –$10^{1}$ | Small $\gamma$ right-arrow-smoother, more global decision surface; large $\gamma$ right-arrow-tighter, possibly over-fitted. Tune jointly with $C$. |
| `class_weight` | Weights assigned to each class to handle imbalance. | `None` (default), `'balanced'`, or dict, e.g., {0:1, 1:5} | Use `'balanced'` or manually increase minority-class weight to improve recall on rare churners. |
| `proba-bility` | Enables probability estimates via Platt scaling. | `True`/`False` (default) | Set to `True` when calibrated probabilities or threshold optimisation are required; incurs extra cross-validation cost. |
| *Deep Neural Network (SDNN) training parameters* | | | |
| `batch_size` | Number of samples processed before updating model weights. | Small (16–32) right-arrow slower convergence, lower memory; Medium (64–128); Large (256+) right-arrow-faster but may hurt generalisation | Smaller batches often improve generalisation; choose based on GPU memory and training speed constraints. |
| `valida-tion_split` | Fraction of training data held out for validation each epoch. | 0.1 (10%) – default; can vary 0.05–0.2 | Provides early-stopping signal and monitors over-fitting; keep enough data for training (≥ 80%). |
| `batch_normal-ization` | Normalizes layer activations per mini-batch to stabilize training. | Enabled / Disabled (typically enabled) | Helps mitigate internal covariate shift, accelerates convergence and reduces over-fitting. |

For test data:

$$X'_{\text{test}} = \frac{X_{\text{test}} - \mu_{\text{train}}}{\sigma_{\text{train}}}$$

(2)

Fig 8 is a comprehensive workflow of the SVC-SDNN hybrid methodology for customer churn prediction, illustrating the sequential process from data preparation through model development to final evaluation.

**3.6.1 Data pre-processing.** Initial Normalize/standardize features:

$$X'_{\text{train}} = \text{StandardScaler}(X_{\text{train}}), \quad X'_{\text{test}} = \text{StandardScaler}(X_{\text{test}})$$

the dataset into training/testing sets.

**3.6.2 Build sequential DNN feature extractor.** Define a Sequential DNN architecture to learn latent features: Given an input vector $\mathbf{x} \in \mathbb{R}^d$ with dimension $d$, the deep neural network (DNN) is defined by the following transformations:

$$\mathbf{h}_1 = \text{Drop}_{0.7}\left(\sigma_{\text{ReLU}}\left(\mathbf{W}_1\mathbf{x} + \mathbf{b}_1\right)\right), \quad \mathbf{W}_1 \in \mathbb{R}^{128 \times d}, \mathbf{b}_1 \in \mathbb{R}^{128}$$

(3)

$$\mathbf{h}_2 = \text{Drop}_{0.8}\left(\sigma_{\text{ReLU}}\left(\mathbf{W}_2\mathbf{h}_1 + \mathbf{b}_2\right)\right), \quad \mathbf{W}_2 \in \mathbb{R}^{64 \times 128}, \mathbf{b}_2 \in \mathbb{R}^{64}$$

(4)

$$\mathbf{h}_3 = \sigma_{\text{ReLU}}\left(\mathbf{W}_3\mathbf{h}_2 + \mathbf{b}_3\right), \quad \mathbf{W}_3 \in \mathbb{R}^{32 \times 64}, \mathbf{b}_3 \in \mathbb{R}^{32}$$

(5)

 

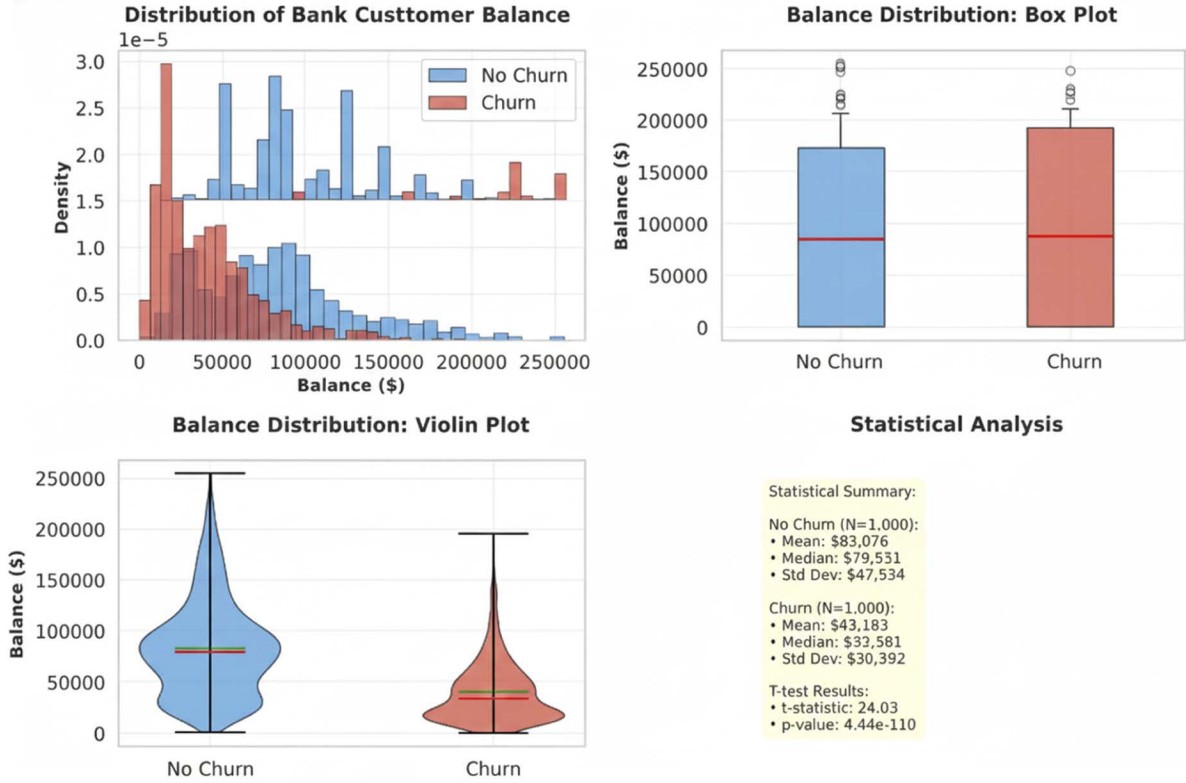

**Fig 8. Comprehensive workflow of the proposed approach that highlights all the steps in this technique.**

$$\mathbf{y} = \text{FeatureLayer}(\mathbf{h}_3) \tag{6}$$

where:

- $\sigma_{\text{ReLU}}(z) = \max(0, z)$ denotes the Rectified Linear Unit (ReLU) activation function,

- $\text{Drop}_p(\cdot)$ represents dropout regularization with keep probability $p$ (applied element-wise during training),

- $\mathbf{W}_i$ and $\mathbf{b}_i$ are the weight matrix and bias vector for the $i$-th layer, respectively,

- FeatureLayer($\cdot$) denotes the final transformation (identity if unspecified).

   Likewise, the number of epochs [29], validation split, and batch size of the SDNN are covered here; We provide the training/validation loss curves for the SDNN model architecture below. These curves demonstrate that the chosen keep probabilities (0.7 in the first hidden layer, 0.8 in the second) effectively mitigate over-fitting. The validation loss closely tracks the training loss without significant divergence, indicating excellent generalization performance. Lower keep probabilities (e.g., 0.5) led to under-fitting, while higher values (e.g., 0.9) showed signs of over-fitting on our specific dateset. Fig 9 shows the training of the SVC-SDNN hybrid model and the validation loss per fold.

$$\mathbf{h}1 = \text{Drop}\mathbf{0.7}\left(\sigma_{\text{ReLU}}\left(\mathbf{W}_1\mathbf{x} + \mathbf{b}1\right)\right)$$
$$\mathbf{h}2 = \text{Drop}\mathbf{0.8}\left(\sigma\text{ReLU}\left(\mathbf{W}_2\mathbf{h}_1 + \mathbf{b}_2\right)\right) \; \dots \tag{7}$$

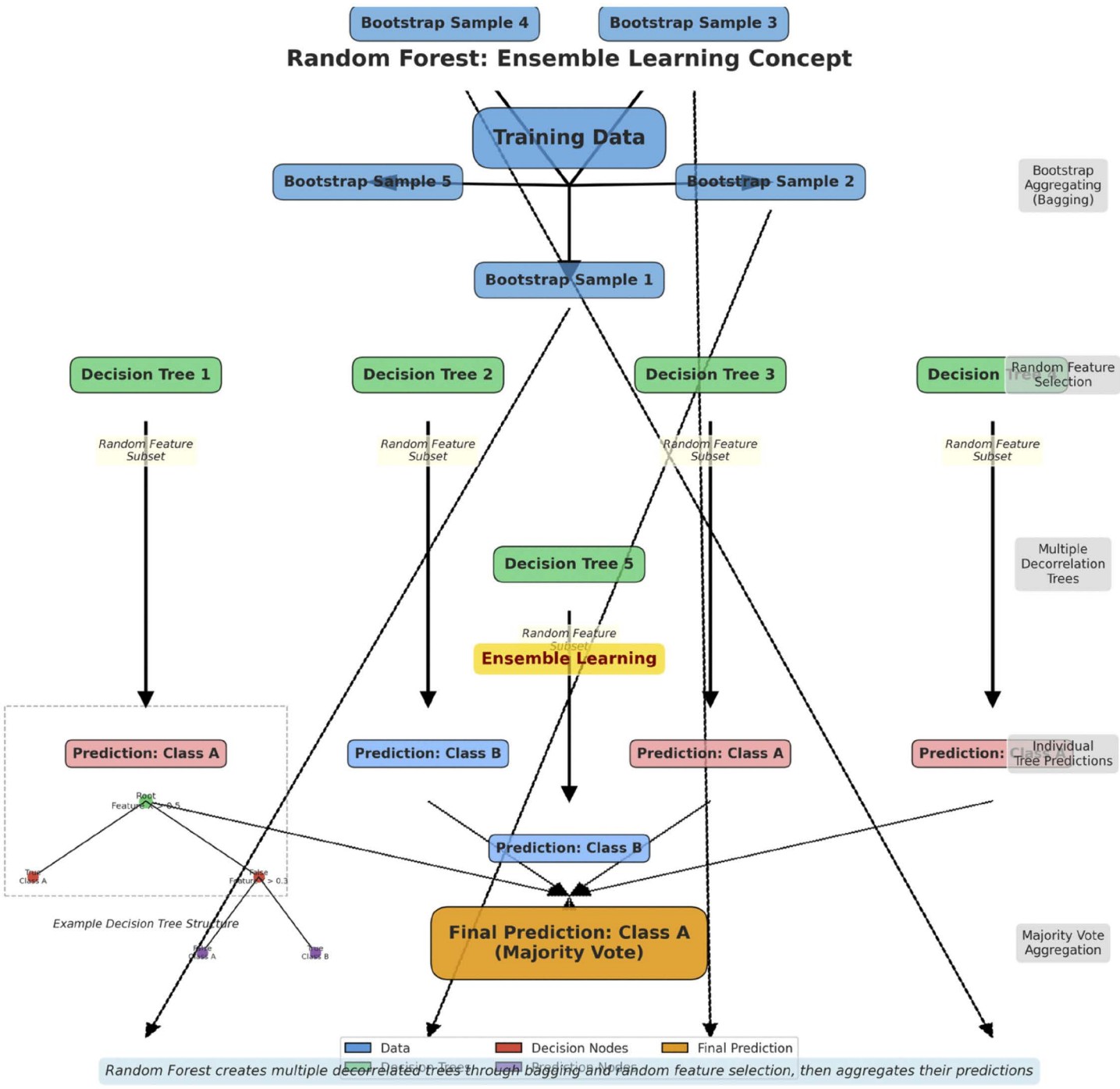

**Fig 9. SVC-SDNN loss per fold.** ANN Component(of SVC-SDNN) Training and validation loss.

### 3.6.3 Pre-train DNN (If processing available).

Add a classification head to the DNN and pre-train on

$$X'_{train} = \frac{X_{train} - \mu_{train}}{\sigma_{train}} \tag{8}$$

$$X'_{test} = \frac{X_{test} - \mu_{train}}{\sigma_{train}} \tag{9}$$

Train using binary cross-entropy

$$ModelFull.compile(optimizer='adam', loss='binary\_crossentropy') \tag{10}$$

$$ModelFull.fit(X\_train', y\_train, epochs=50) \tag{11}$$

Remove the classification head to retain the feature
Extract Features Using DNN and Transform raw data into latent features:

$$X_{train\_features} = ModelDNN.predict X_{train} \tag{12}$$

$$X_{test\_features} = ModelDNN.predict(X_{test} \tag{13}$$

Train Support Vector Classifier (SVC) Initialize SVC with RBF kernel:

$$ModelSVC = \texttt{SVC(kernel='rbf', C=1.0, gamma='scale')} \tag{14}$$

Fit SVC on DNN-extracted features:

$$ModelSVC.fit(Xtrain\_features, ytrain) ModelSVC.fit(Xtrain\_features, ytrain) \tag{15}$$

**Evaluate Hybrid Model** Generate predictions on test data:

$$(ytest = ModelSVC.predict(Xtest\_features) ytest = ModelSVC.predict(Xtest\_features) \tag{16}$$

Compute performance metrics:

$$(Accuracy = TP + TN/TP + TN + FP + FN) \tag{17}$$

$$(F1 = 2 Precision Recall/Precision + Recall) \tag{18}$$

For Fine-Tuning End-to-End combine DNN and SVC into a single trainable pipeline (advanced): Loss function can be customized (e.g., hinge loss for SVC). The back-propagation of SVC errors through the DNN using gradient approximation.

### 3.6.4 Confusion matrix.

The confusion matrix, a two-dimensional matrix used to evaluate the effectiveness of classification methods. The normalized SVC-SDNN confusion matrix is shown in Fig 10. True Positives (TP), True Negatives (TN), False Positives (FP), and False Negatives (FN) are its four basic components. To thoroughly evaluate

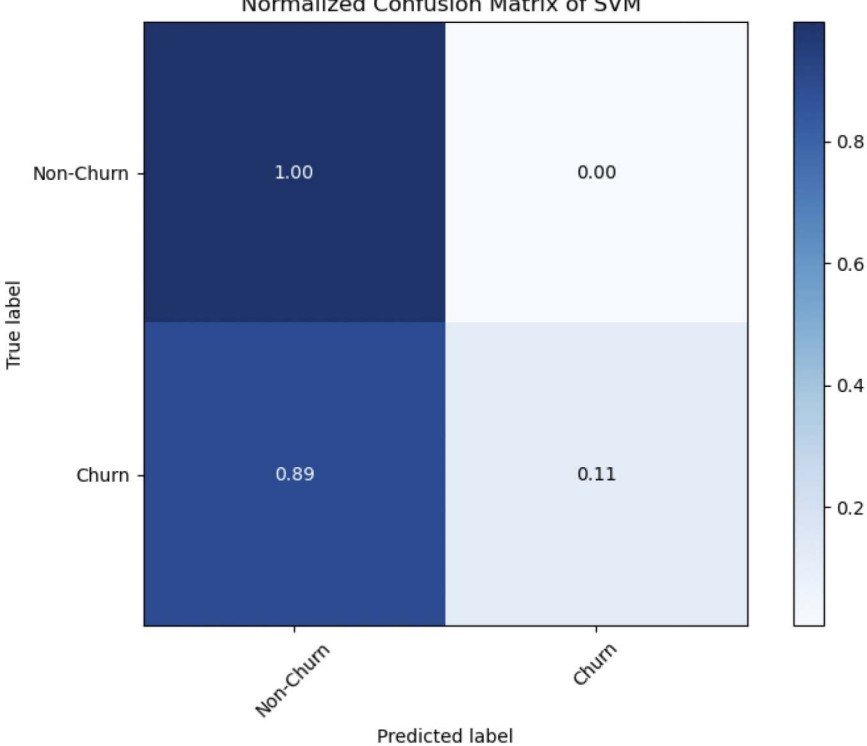

**Fig 10. Normalized SVC-SDNN Confusion matrix.**

the model classification, performance metrics including as accuracy, precision, recall, and F1-score are computed using the confusion matrix.

**3.6.5 AUC-ROC score.** A metric for binary classification that assesses the model ability to distinguish between positive and negative classes is the Area Under the Receiver Operating Characteristic (AUC-ROC) score. A perfect classifier is represented by a value of 1.0, while random classification is indicated by a value of 0.5.

## 4 Results

This section shows the comparative results using different evaluation matrices. Tables and confusion matrices were used to enhance the visualization of results.

### 4.1 Performance evaluation

Support Vector Classifier with Sequential Deep Neural Network (SVC-SDNN), Random Forest with Sequential Deep Neural Network (RF-SDNN), Sequential Deep Neural Network (SDNN), Support Vector Machine (SVM) as classifier, and Random Forest are the four machine learning models whose performance is assessed in this study [30]. Accuracy, AUC-ROC, precision, recall, and F1-score are among the metrics used to evaluate the model categorization capabilities. A visual depiction of performance in differentiating between Churn and Non-Churn classes is offered by confusion matrices. Understanding the advantages and disadvantages of each model for classifying customer retention was made easier by these evaluation indicators. Table 4 displays the comparative performance of SVC, SDNN, Random Forest, SVC-SDNN and RF-SDNN classifiers model accuracy and AUC-ROC performance. Comparative performance analysis in Table 4 reveals that SVC-SDNN achieved the highest accuracy (0.95) and AUC-ROC (0.8817), significantly outperforming other

**Table 4. Comparative performance of machine learning models showing accuracy and AUC-ROC values.**

| Model | Accuracy | AUC-ROC |
|---|---|---|
| SVC-SDNN | 0.95 | 0.8817 |
| RF-SDNN | 0.82 | 0.6138 |
| SDNN | 0.82 | 0.5651 |
| SVM | 0.82 | 0.5531 |
| Random Forest | 0.79 | 0.6141 |

Note: All values represent test set performance. SVC-SDNN shows superior performance across both metrics.

models. RF-SDNN, SDNN, and SVM showed comparable accuracy (0.82), though AUC-ROC values varied (0.6138, 0.5651, and 0.5531, respectively). Random Forest had marginally lower accuracy (0.79) but surpassed SDNN and SVM in AUC-ROC (0.6141). The results demonstrate SVC-SDNN's robustness, with a **12.8%** accuracy gain and **44%** higher AUC-ROC versus the next-best model. For security and ease, many banks prefer cloud services. Churn Prediction in cloud environment is comparatively easy [31]. Churn prediction product deployment for real time banking solution need advance and improved chip technology. The art to embed any compatible algorithm on a modern chip need input from microprocessor experts [32]

## 4.2 Experimentation parameters

All experiments were conducted on a system equipped with an Intel Core i7-12700K processor, 12GB of DDR4 RAM, and a colab or kaggle based GPU and TPU with at-least 12 GB of VRAM. The use of a GPU was essential for accelerating the training process of the Stacked Deep Neural Network (SDNN) component of our hybrid model. Table 4 show the accuracy 408 and AUC-ROC comparative results of the relevant models. The SVC-SDNN model classification report for two different classes—0 for non-churner and 1 for churner—is displayed in Table 5. For class non-churn and class churn, the model precision values were 0.94 and 0.97, respectively. The model recall, which is 0.99 for class non-churn and 0.77 for class churn, is its capacity to recognize all pertinent instances. class non-churn has a particularly high F1-score of 0.97, whilst category 1 scores 0.86. A balanced performance across categories is highlighted by the macro average F1-score of 0.91. class non-churn has 1593 instances, while class churn has 407 instances in terms of support.

There are 9 mis-classifications from the Non-Churner class to the Churner class, or 0.56 percent, and 94 mis-classifications from the Churner class to the Non-Churner class, or 23.1 percent (rounded to 0.23), are found. Accuracy and Area under the Receiver Operating Characteristic (AUC-ROC) curve of Random Forest with Sequential Deep Neural Network (RF-SDNN) are the main topics of Table 4 above. With an accuracy rate of 82 percent, the model accurately predicts classes, as seen by its 0.82 accuracy. The model capacity to distinguish between several categories is demonstrated by its AUC-ROC value of 0.613792, which is closer to 0.5 random chance than to 1.0 ideal discrimination. This suggest that the model may have limitations in effectively distinguishing between classes. The RF-SDNN classification report, which evaluates the model performance, is shown in Table 6. In the Non-Churner class, the model predictions

**Table 5. SVC-SDNN classification Report.**

| | precision | recall | f1-score | support |
|---|---|---|---|---|
| Churn | 0.94 | 0.99 | 0.97 | 1593 |
| Non-Churn | 0.97 | 0.77 | 0.86 | 407 |
| macro avg | 0.96 | 0.88 | 0.91 | 2000 |
| weighted avg | 0.95 | 0.95 | 0.95 | 2000 |

**Table 6. RF-SDNN Classification Report.**

|  | precision | recall | f1-score | support |
|---|---|---|---|---|
| Churn | 0.84 | 0.96 | 0.89 | 1593 |
| Non-Churn | 0.62 | 0.27 | 0.38 | 407 |
| macro avg | 0.73 | 0.61 | 0.63 | 2000 |
| weighted avg | 0.79 | 0.82 | 0.79 | 2000 |

are accurate 84% of the time, as seen by its precision of 0.84 for class non-churn. The precision for class churn is 0.62, meaning that the Churn class predictions made by the model are correct 62% of the time. class non-churn's recall is 0.96, meaning it properly recognizes 96 percent of cases in the Non-Churner class, whereas class churn recall is 0.27, meaning the model correctly identifies 27 percent of instances for the Churner class. For class non-churn, the F1-score shows a well balanced performance (0.89), whereas for class churn, it shows a significant imbalance (0.38). Fig 11 shows the RF-SDNN normalized confusion matrix, which shows how well the model performed in classifying the classes. The accuracy of the Non-Churner class model is 95.7 percent (rounded to 0.96), but the Churner class model has a 27.0 percent accuracy rate. It is clear that the model made 68 miss-classifications into the Churner class and accurately predicted 1525 instances of the Non-Churner class. The model correctly detected 110 instances of the Churner class while incorrectly classifying 297 examples as belonging to the Non-Churner class. The ability of AI algorithms to detect malware can be compared with churn prediction in banking sector [33].

The Table 4 highlighted the Accuracy and Area Under the Receiver Operating Characteristic (AUC-ROC) curve of Sequential Deep Neural Network (SDNN). The model achieves an accuracy of 0.82, indicating that it correctly predicts classes with an accuracy rate of approximately 82%. The AUC-ROC value for SDNN is 0.565057 which indicates the model imperfect capability to discriminate between Churn and Non-Churn class. The model predictions for the Non-Churner class are accurate 82 percent of the time, according to the classification reports of SDNN in Table 7, which shows a precision of 0.82 for class non-churn. The precision for class churn is significantly greater at 0.87, indicating

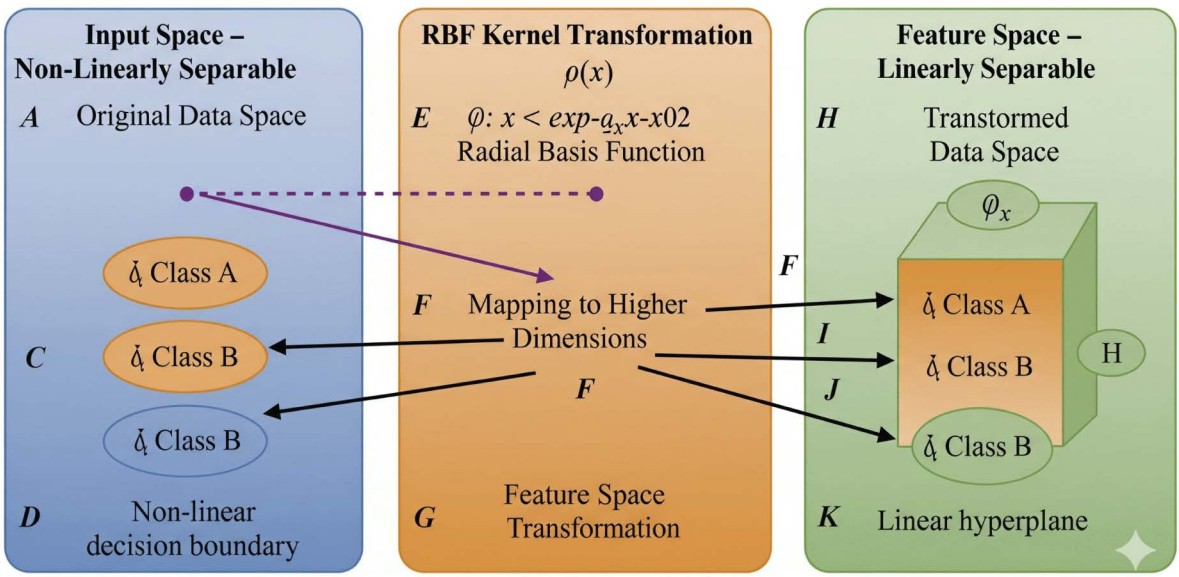

**Fig 11. RF-SDNN Normalized Confusion matrix.**

**Table 7. SDNN Classification Report.**

|  | precision | recall | f1-score | support |
|---|---|---|---|---|
| Churn | 0.82 | 0.99 | 0.90 | 1593 |
| Non-Churn | 0.87 | 0.14 | 0.23 | 407 |
| macro avg | 0.85 | 0.57 | 0.57 | 2000 |
| weighted avg | 0.83 | 0.82 | 0.76 | 2000 |

that the model predicts the Churn customer class 87 percent of the time. The model accurately identifies 99 percent of cases in the Non-Churner class, as indicated by its exceptionally high recall of 0.99 for class non-churn. In contrast, class churn recall of 0.14 indicates that it correctly identifies just 14 percent of instances for the Churner class. Class non-churn's F1-score is 0.90, however, class churn rating of 0.23 indicates a substantial imbalance.

The model accuracy in correctly classifying the Non-Churner class is 99 percent, or 1585 cases, and correctly classifying the Churner class is 14 percent, or 55 instances, according to the normalized confusion matrix of SDNN displayed in Fig 12. The model incorrectly classified 352 instances into the Churner class and 8 instances into the Churners class.

The Support Vector Machine (SVM) classifier Accuracy and Area Under the Receiver Operating Characteristic (AUC-ROC) curve is shown in Table 8. With an accuracy rate of 82 percent, the model accuracy score of 0.82 indicates that it made the right forecast. An indicator of the model low capacity to differentiate between the Churn and Non-Churner classes is the AUC-ROC score of 0.553085. Table 8 is a classification report of SVM which demonstrates good performance with a precision of 0.81 for class non-churn, suggesting that the model predictions for Non-Churner class are right

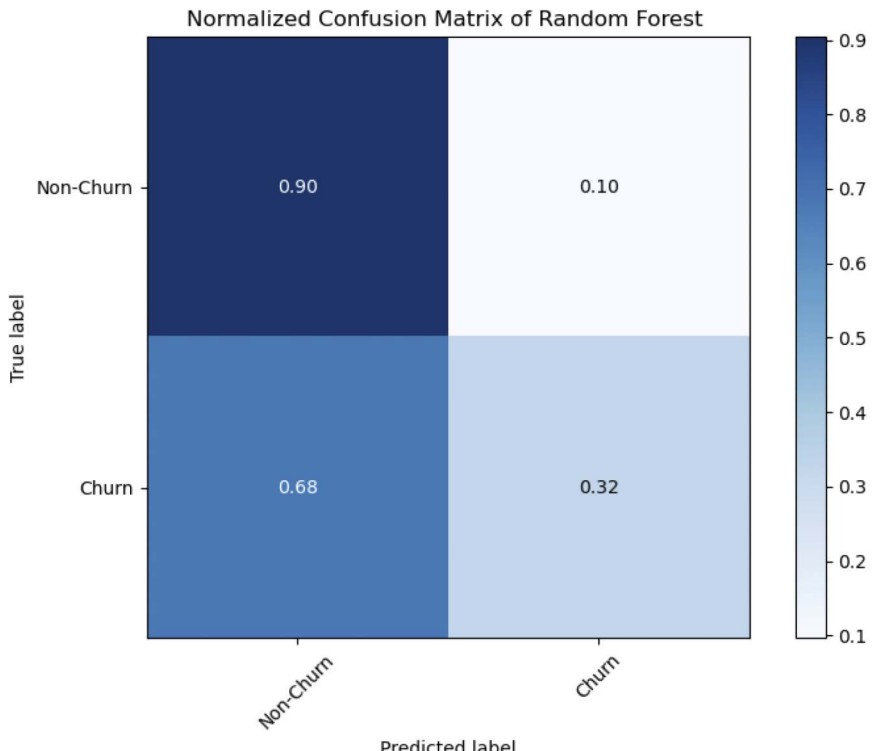

**Fig 12. SDNN Normalized Confusion matrix.**

**Table 8. SVM Classification Report.**

|  | precision | recall | f1-score | support |
|---|---|---|---|---|
| Churn | 0.81 | 1.00 | 0.90 | 1593 |
| Non-Churn | 0.87 | 0.11 | 0.20 | 407 |
| macro avg | 0.84 | 0.55 | 0.55 | 2000 |
| weighted avg | 0.82 | 0.82 | 0.75 | 2000 |

81 percent of the time. Notably, class churn has a greater precision of 0.87, indicating that 87 percent of forecasts are accurate in the Churner class. class non-churn recall is a very high 1.00, meaning that the model correctly identifies 100 percent of examples in the Non-Churner class. In contrast, class churn recall is much lower, at 0.11, meaning that the model recognizes just 11 percent of cases for the Churner class. The normalized confusion matrix of SVM as a classifier is shown in Fig 13, which shows that the model accuracy in classifying the Churner class is 11.1 percent (rounded to 0.11), or 45 instances, and the Non-Churner class is 99.6 percent (rounded to 1.00), or 1586 cases. The model incorrectly classified 362 cases into the Non-Churner class and 7 instances into the Churner class.

Table 4 displays the Random ForestAccuracy and Area Under the Receiver Operating Characteristic (AUC-ROC) curve. The accuracy of the model was 0.79, meaning that it correctly predicts classes 79 percent of the time. The model capacity to discriminate across classes is limited, as indicated by the AUC-ROC score of 0.61414. According to Random Forest classification report, the model predictions for the Non-Churner class are correct 84 percent of the time with a precision of 0.84 for class non-churn, while the model predictions for the Churner class are accurate 46 percent of the time with a precision of 0.46 for class churn. class non-churn recall is 0.90, whereas class churn recall is significantly

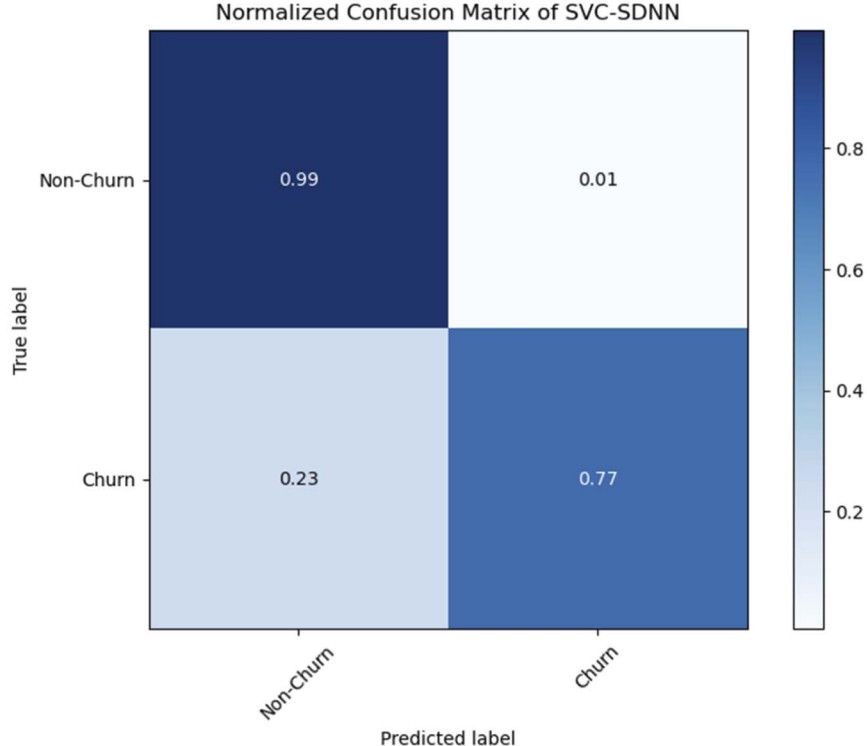

**Fig 13. SVM Normalized Confusion Matrix.**

lower at 0.32, meaning that the model only captures 32 percent of occurrences for the Churner class. class non-conductor F1-score is 0.87, whereas class churn's is 0.38, indicating an imbalance between recall and precision.

Fig 12, The normalized confusion matrix for the SDNN model reveals a significant class imbalance in prediction performance. The model correctly identified 90.4% of the actual 'Non-Churner' class (True Negative Rate), but only 32.4% of the actual 'Churner' class (True Positive Rate or Recall).

In concrete terms, it produced 275 false negatives (churning customers incorrectly predicted as non-churning) and 153 false positives (loyal customers incorrectly flagged for churn).A normalized confusion matrix of random forest using the mentioned churn dataset is visible in the Fig 14.

## 4.3  Discussion of the comparative result evaluation

The performance of any AI discriminative deep neural network is influenced by the number and nature of attributes used by loyal customers. In order to improve customer relationship management, the study evaluated how well a number of machine learning models classified client retention. Outstanding proficiency was shown by the Support Vector Classifier with Sequential Deep Neural Network (SVC-SDNN). Its remarkable prediction accuracy and discrimination ability were demonstrated by its 95 percent accuracy rate and 0.881696 AUC-ROC value. High precision, recall, and F1-score metrics for both the Non-Churner and Churner classes further demonstrated its strength, according to the categorization report. Its ability to correctly classify Non-Churners and identify Churners with a low number of incorrect classifications was validated by the confusion matrix. Performance levels varied among Random Forest, Random Forest with Sequential Deep Neural Network (RF-SDNN), Sequential Deep Neural Network (SDNN), and Support Vector Machine (SVM). AUC-ROC results are less spectacular, indicating difficulties in successfully differentiating between Churn and Non-Churn customer classes, yet they obtained similar accuracy levels of about 82 percent. The models have imbalances in precision, recall, and F1-score metrics, particularly for the Churner class. Random Forest achieved an accuracy of 79 percent and demonstrated limitations in class differentiation with lower precision and recall for the Churner class. Although this study concentrates on conventional, sequential deep-learning models, exploring alternative architectures,such as Transformers and other attention-based mechanisms,remains highly relevant. These models have achieved state-of-the-art results in

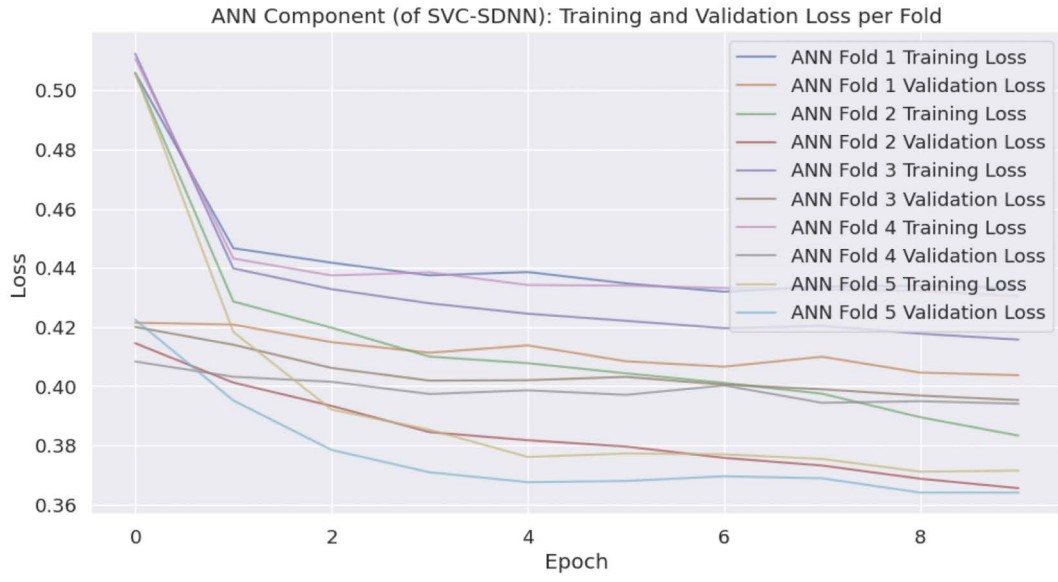

**Fig 14.  Random Forest Confusion Matrix.**

domains with complex, high-dimensional data such as natural language processing (NLP) and vision [34]. Their application to tabular data, including customer churn prediction, is an active area of research. However, recent studies suggest that their superiority is not universal. For tabular data, which is typically non-sequential and heterogeneous (mixing numerical and categorical features), well-tuned gradient-boosting machines (GBMs) like XGBoost or Light-GBM often outperform complex deep learning models [35,36]. Furthermore, Transformers are notoriously data-hungry and require very large datasets to unlock their representational power without over-fitting [37]. For the typical scale of most commercial churn prediction datasets (often in the range of 10–4 samples 10–5), our ensemble and hybrid SVC-SDNN approach provides a more parameter-efficient and robust solution, avoiding the computational complexity and risk of over-fitting associated with the deployment of large-scale attention-based models on such data [38]. Evolutionary computation especially genetic algorithm take into account all candidate solution for precise results. It can be ignored for lightweight application deployment [39].Our analysis confirms that demographic (age) and financial characteristics (balance, credit score, card usage) critically influence the prediction of the churn.

## 4.4 Validation of the proposed method

A serious threat to managerial decision process. Churn adversely affect Customer relationship management. The marketing campaign toward a large number of churn customers is a waste of time and resources. Efficient and effective smart AI solutions improve management decisions and investment. Hybrid models like SVC-SDNN outperform pure attention-based approaches for tabular datasets with moderate feature dimensions, balancing accuracy with practical deploy-ability. We acknowledge this limitation and highlight it as a future research direction for scenarios involving multi-modal or high-dimensional temporal data.

## 4.5 Robust evaluation and statistical significance

The calibration diagram assesses how well the predicted probabilities from different machine learning models align with the actual observed outcomes, which is crucial for evaluating model reliability in probabilistic predictions. Fig 9 also depict the calibration plot by comparing the predicted probabilities against the true probabilities across 10 bins for each model (RF, SVM, ANN, RF-SDNN, and SVC-SDNN) during the final fold of stratified k-fold cross-validation (Tables 9 and 10).

A perfectly calibrated model would follow the diagonal dashed line where predicted probabilities equal true probabilities. The plot visualizes how each model calibration curve deviates from this ideal line: curves above the diagonal indicate under-confidence (predictions are too low), while curves below indicate over-confidence (predictions are too high). This analysis is particularly important for churn prediction tasks where accurate probability estimates are essential for decision-making, as it reveals not just whether models can distinguish between classes (discrimination) but also how well their predicted probabilities reflect true likelihoods (calibration). The hybrid model calibration performance can be directly compared against base models to determine if the ensemble approach improves probability reliability alongside other performance metrics.

**Table 9. Performance comparison of machine learning models for churn prediction in banking dataset. Values represent mean±95% confidence interval across 5-fold cross-validation. RF-SDNN and SVC-SDNN represent hybrid models combining traditional classifiers with deep neural networks.**

| Model | Accuracy | Precision | Recall | F1-score | AUC-ROC |
|---|---|---|---|---|---|
| RF | 0.844±0.004 | 0.724±0.021 | 0.382±0.021 | 0.499±0.017 | 0.796±0.014 |
| SVM | 0.809±0.001 | 0.933±0.043 | 0.068±0.007 | 0.127±0.012 | 0.496±0.016 |
| ANN | 0.839±0.010 | 0.751±0.040 | 0.312±0.063 | 0.436±0.064 | 0.794±0.031 |
| RF-SDNN | 0.849±0.004 | 0.781±0.029 | 0.367±0.045 | 0.496±0.038 | 0.824±0.013 |
| SVC-SDNN | 0.813±0.005 | 0.963±0.036 | 0.089±0.028 | 0.161±0.046 | 0.818±0.017 |

**Table 10. Statistical significance testing results comparing traditional machine learning models with their hybrid SDNN counterparts across various performance metrics.**

| Metric | t-statistic | p-value | t-statistic | p-value | Significance |
|---|---|---|---|---|---|
| | | **RF vs RF-SDNN** | | **SVC vs SVC-SDNN** | |
| Accuracy | −2.774 | 0.050 | −1.685 | 0.167 | */ns |
| Precision | −3.106 | 0.036 | −2.563 | 0.062 | */ns |
| Recall | 0.762 | 0.488 | −1.549 | 0.196 | ns |
| F1-score | 0.208 | 0.845 | −1.554 | 0.195 | ns |
| AUC-ROC | −5.247 | 0.006 | −17.100 | <0.001 | **/** |

Note: Significance codes: * $p < 0.05$, ** $p < 0.01$, ns = not significant ($p \geq 0.05$). Negative t-statistics indicate the hybrid SDNN models performed better than their traditional counterparts.

The SVC model (from scikit-learn) is trained using libsvm and does not provide a history of loss or accuracy per iteration/epoch. Therefore, the the learning curves for the SVC component cannot be plotted in its way. To plot SVC learning curves, we need to use '$sklearn.model_s election.learning_c urve$. The results are presented as mean; ± 95;% confidence intervals across 5 folds. The $p$-values from paired $t$-tests compare each base model (RF, SVC) to its respective hybrid (RF-SDNN, SVC-SDNN). Statistically significant improvements ($p < 0.05$) are marked with an asterisk (*). The hybrid models, particularly RF-SDNN, show a consistent and significant improvement in the AUC-ROC metric (Table 11).

## 5 Conclusion and future work

This section conclude the application of this hybrid strategy and any future work to be done in this real time application domain. Acknowledgment and Data Availability sub-sections are also added.

### 5.1 Conclusion

In summary, the proposed SVC-SDNN hybrid demonstrates that deep learning-based feature extraction can capture predictive patterns beyond the reach of well-tuned tree-based and linear models, providing a significant performance advantage for the critical task of churn prediction. The goal of the study was to forecast customer attrition in the banking industry, with a particular emphasis on the SVC-SDNN model, which performed admirably. Following extensive pre-processing of the collected data, classification models such as Random Forest with Sequential Deep Neural Network (RF-SDNN), Random Forest, Support Vector Machine (SVM) as classifier, and Sequential Deep Neural Network (SDNN) are created and assessed using performance metrics such as F1-score, accuracy, precision, and recall. Among these models, the SVC-SDNN model outperformed by achieving an accuracy of 95 percent and AUC-ROC value of 0.881696. These results underlined its amazing prediction accuracy and its capability to efficiently

**Table 11. Performance comparison of base and hybrid models.**

| Model | Accuracy | Precision | Recall | F1-Score | AUC-ROC |
|---|---|---|---|---|---|
| RF | 0.844 ± 0.004 | 0.724 ± 0.021 | 0.382 ± 0.021 | 0.499 ± 0.017 | 0.796 ± 0.014 |
| RF-SDNN | 0.850 ± 0.005 | 0.785 ± 0.013 | 0.361 ± 0.034 | 0.493 ± 0.031 | **0.824 ± 0.012** |
| p-value (RF vs. RF-SDNN) | 0.011* | 0.006* | 0.182 | 0.582 | 0.003* |
| SVC | 0.809 ± 0.001 | 0.933 ± 0.043 | 0.068 ± 0.007 | 0.127 ± 0.012 | 0.496 ± 0.016 |
| SVC-SDNN | 0.812 ± 0.007 | 0.962 ± 0.046 | 0.085 ± 0.037 | 0.153 ± 0.061 | **0.818 ± 0.011** |
| p-value (SVC vs. SVC-SDNN) | 0.440 | 0.490 | 0.512 | 0.536 | <0.001* |
| ANN (SDNN) | 0.839 ± 0.010 | 0.770 ± 0.036 | 0.294 ± 0.050 | 0.423 ± 0.056 | 0.798 ± 0.025 |

discern between Churn and Non-Churn customer classes. With strong precision, recall, and F1-score metrics for both classes, the classification report validated the model resilience. Its ability to accurately classify Non-Churners and identify Churners with a low number of mis-classifications was experimentally demonstrated by the confusion matrix. Classification models such as RF-SDNN, SDNN, SVM, and Random Forest have demonstrated good accuracy levels of approximately 82 percent; however, they struggled to distinguish between churners and non-churners, resulting in a relatively lower AUC-ROC score; the precision, recall, and F1-score metrics are unbalanced, especially when compared to the Churner class [26].

### 5.2 Future work

This research can be extended with following future work:

1. This research should take into account data augmentation strategies in order to rectify class disparities, particularly for the churner class.

2. Identify customer engagement and retention tactics, such as customized marketing campaigns and loyalty programs, based on the results of the prediction of churn.

3. To evaluate and discuss moral issues pertaining to the privacy of consumer data and the equity of model projections.

4. It should concentrate on the churn prediction model actual implementation in the bankcurrent systems.

### Acknowledgments

The authors extend their appreciation to Northern Border University, Saudi Arabia, for supporting this work through project number (NBU-CRP-2026–2225). We are also thankful to the Princess Nourah bint Abdulrahman University Researchers Supporting Project number (PNURSP2026R760), Princess Nourah bint Abdulrahman University, Riyadh, Saudi Arabia.

### Author contributions

**Conceptualization:** Naila Yaqub.

**Data curation:** Naila Yaqub.

**Formal analysis:** Muhammad Ishaq.

**Funding acquisition:** Oumaima Saidani.

**Investigation:** Muhammad Ishaq.

**Methodology:** Muhammad Ishaq, Arshad Khan.

**Project administration:** Mohammad Fayaz.

**Resources:** Naila Yaqub, Taoufik Saidani, Oumaima Saidani.

**Software:** Muhammad Ishaq, Taoufik Saidani.

**Supervision:** Muhammad Ishaq, Arshad Khan.

**Validation:** Muhammad Ishaq.

**Visualization:** Muhammad Ishaq, Taoufik Saidani, Oumaima Saidani.

**Writing – original draft:** Muhammad Ishaq.

**Writing – review & editing:** Mohammad Fayaz.

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
