## [Editor Report · Decision Letter 0]

2 Jan 2024

PONE-D-23-42083

CLASSIFICATION OF CUSTOMER RETENTION USING HYBRID SVC-SDNN TO ENHANCE CUSTOMER RELATIONSHIP MANAGEMENT

PLOS ONE

Dear Dr. Ishaq,

Thank you for submitting your manuscript to PLOS ONE. After careful consideration, we have decided that your manuscript does not meet our criteria for publication and must therefore be rejected.

I am sorry that we cannot be more positive on this occasion, but hope that you appreciate the reasons for this decision.

Kind regards,

Asadullah Shaikh, Ph.D.

Academic Editor

PLOS ONE

Additional Editor Comments:

This paper does not provide a solid method for Accuracy and AUC-ROC of SVC-SDN

In addition, the paper structure is not well managed. Also, there is no comparison in the related work section.

The references are quite old.

The accuracy claim is too high but its not well explained

- - - - -

---

## [Author Response · Author response to Decision Letter 1]

29 Feb 2024

Comments 1:

After careful consideration, we have decided that your manuscript does not meet our criteria for publication and must therefore be rejected.

Response 1:

I was busy reviewing your already-published relevant manuscripts on Google Scholar.

Before Submission and after receiving an email regarding the decision, I thoroughly reviewed the recent relevant articles in PLOS ONE.

The description of the models is not schematic or superficial. Most of the important concepts are mentioned on a need-by-need basis (as required).

Many of PLOS ONE's and your recent articles follow the same structured approach.

Comment 2:

I am sorry that we cannot be more positive on this occasion, but hope that you appreciate the reasons for this decision.

Response 2:

There is a comprehensive presentation of the problem and its peculiarities. Could you please elaborate on the in-depth description of your rejection decision?

Comments 3:

This paper does not provide a solid method for Accuracy and AUC-ROC of SVC-SDN.

Response 3:

The method for Accuracy and AUC-ROC of SVC-SDN is exactly the same as we find it in recent relevant literature. The problem is also characterized, and the evaluation of each approach is given with results.

Comment 4:

In addition, the paper structure is not well managed.

Response 4:

PLOSONE initially allow articles submission in open format. After acceptance we are bound to fully adopt the Journal standard structure.

Comment 5:

Also, there is no comparison in the related work section.

Response 5:

SVC-SDN is compared with several other recent approaches in the related work as well as results and experimental section.

Comments 6:

The references are quite old.

Response 6:

What can we do with old references of scientific importance? In my opinion we must include it. There are plenty of recent references.

I even shared the notebook on github and the churn dataset links.

There are more recent variants in the latest press, and it will take time to work them out as a future work.

I personally find many typos and grammar mistakes in published articles and even in your Decision Email.

The Sign of a true researcher is a rational and objective mindset. Many Editors/Reviewers follow prejudices.

Comments 7:

I am sorry that we cannot be more positive on this occasion, but hope that you appreciate the reasons for this decision.

Response 7:

Based upon the above point-by-point response of your editorial opinion, we are quite hopeful to reconsider the same article for onward processing.

Comments 8:

The accuracy claim is too high but its not well explained.

Response 8:

As a programmer Its really time consuming and tedious task to improve the model performance over a dataset. Several layers are added and removed for better results.

Layer in any deep NN represent either a neuron or a network of motor, Sensory and associative neurons.

---

## [Decision Letter · Decision Letter 1]

20 Feb 2025

PONE-D-23-42083R1CLASSIFICATION OF CUSTOMER RETENTION USING HYBRID SVC-SDNN TO ENHANCE CUSTOMER RELATIONSHIP MANAGEMENTPLOS ONE

Dear Dr. Ishaq,

Thank you for submitting your manuscript to PLOS ONE. After careful consideration, we feel that it has merit but does not fully meet PLOS ONE’s publication criteria as it currently stands. Therefore, we invite you to submit a revised version of the manuscript that addresses the points raised during the review process.

We look forward to receiving your revised manuscript.

Kind regards,

Sajid Anwar, Ph.D

Academic Editor

PLOS ONE

● A copy of your manuscript showing your changes by either highlighting them or using track changes (uploaded as a supporting information file)

● A clean copy of the edited manuscript (uploaded as the new manuscript file)

Additional Editor Comments (if provided):

Reviewers' comments:

Reviewer's Responses to Questions

**Comments to the Author**

1. If the authors have adequately addressed your comments raised in a previous round of review and you feel that this manuscript is now acceptable for publication, you may indicate that here to bypass the “Comments to the Author” section, enter your conflict of interest statement in the “Confidential to Editor” section, and submit your "Accept" recommendation.

Reviewer #1: (No Response)

Reviewer #2: (No Response)

2. Is the manuscript technically sound, and do the data support the conclusions?

Reviewer #1: Yes

Reviewer #2: Yes

3. Has the statistical analysis been performed appropriately and rigorously? 

Reviewer #1: N/A

Reviewer #2: Yes

4. Have the authors made all data underlying the findings in their manuscript fully available?

Reviewer #1: Yes

Reviewer #2: Yes

5. Is the manuscript presented in an intelligible fashion and written in standard English?

Reviewer #1: (No Response)

Reviewer #2: Yes

6. Review Comments to the Author

Reviewer #1: The paper presents improved churn detection using novel hybrid deep learning algorithms to predict bank customers’ retention and identify factors that influence better customer retention. SVC-SDNN model was proposed for churn prediction. The proposed model was compared to other existing models with the proposed model having a better performance. The performance of the proposed model is encouraging. However, I have the following observations:

1. The claim in second paragraph under section 1.1 should be cited.

2. Sections 1.2 to 1.5 are not needed in a paper. I think those sections are relevant to project/thesis report. If there is anything important at all in those sections, especially, section 1.5 that talks about the structure of the paper should be married to the introduction as the last paragraph.

3. Figure 1 citation is incomplete, it should be a surname, year.

4. Section II should be renamed as Background of the Study and Related Work because Sections 2.1 and 2.2 fall under background

5. In-text citations in Table 1 do not need initials but only surname(s) and year. In addition, the table should provide a succinct summary of the related work and research gaps and not just a repetition of titles and dataset.

6. Remove any line of codes in the paper. Codes can be added as appendix or supplementary if required.

7. The dataset is skewed as the authors have also notified. There is need to balance the dataset to avoid model over/under-fitting.

8. The authors claimed that relevant features were selected. However, what informed the selection of the relevant features or better still the technique for feature selection was not stated.

9. The hybridization of the SVC and SDNN was not clear in the paper. Authors are to provide the details of hybridization to enable reproducibility.

10. Allow a line spacing before and after a figure/table for readability.

11. I suggest you merge Tables 2 to 11 as one so that one can easily see the difference in performance among the models at a glance.

12. The confusion matrices from Figures 9 to 13 can be arranged together or sequentially (may be 3 in a row) for readability and comparison.

13. The structure of the paper is not good enough. It is advisable to follow the journal template.

Reviewer #2: Although, I haven't previously review the paper, However, evaluating the authors response to the reviewers, authors have comprehensively addressed the reviewer comments in the revised version. Still, to improve the quality of the paper, below are certain comments that need to be address

1. The authors should add more technical details about the proposed approach in the abstract, i.e., talking more about about pre processing, feature engineering, or model details.

2. The authors should enlist the key contribution of the proposed approach.

3. The support vector machine and random forest should be moved to the result or methodology section where the proposed approach is elaborated, the related work should only focus on the state-of-the-art literature.

4. Figure 5 seems unnecessary, instead a Table should be added that shows several example from the dateset.

5. There are many deep-learning based approaches published on the topic. What authors motivate to proposed a hybrid approach while not using state-of-the-art transformers or LLMs? Adding a more rationale will further improve the justification of the proposed hybrid approach

5. The authors should further provide details on the implementation of the proposed approach, such as on how SVM is added to the DNN classifier?

7. The author should demonstrate or provide the rationale on how the simple banking system dateset attributes are enough for churn prediction for the generalization of the proposed approach.

8. Also, what motivate the authors for the proposed approach, as there are better alternatives to the proposed approach by employing large language models.

9. In Tables, the the labels 0 and 1 should be replaced with more intuitive one to better understand the results.

10. Summary should be replaced with Discussion and add some insights how the proposed approach help state-of-the-art

11. The authors should add threats to validity section to identify the possible threats in the proposed approach.

7. PLOS authors have the option to publish the peer review history of their article (what does this mean? ). If published, this will include your full peer review and any attached files.

**Do you want your identity to be public for this peer review?** For information about this choice, including consent withdrawal, please see our Privacy Policy .

Reviewer #1: No

Reviewer #2: **Yes:** Javed Ali khan

While revising your submission, please upload your figure files to the Pre-flight Analysis and Conversion Engine (PACE) digital diagnostic tool, https://pacev2.apexcovantage.com/ . PACE helps ensure that figures meet PLOS requirements. To use PACE, you must first register as a user. Registration is free. Then, login and navigate to the UPLOAD tab, where you will find detailed instructions on how to use the tool. If you encounter any issues or have any questions when using PACE, please email PLOS at figures@plos.org . Please note that Supporting Information files do not need this step.

---

## [Author Response · Author response to Decision Letter 2]

23 Mar 2025

Respected Reviewers,

We are really thankful for in depth remarks and comments for further improvement of our article.

The Detail response to reviewers file is uploaded. The clean Manuscript and the Track change Manuscript file are also uploaded.

Regards

---

## [Decision Letter · Decision Letter 2]

26 May 2025

PONE-D-23-42083R2CLASSIFICATION OF CUSTOMER RETENTION USING HYBRID SVC-SDNN TO ENHANCE CUSTOMER RELATIONSHIP MANAGEMENTPLOS ONE

Dear Dr. Ishaq,

Thank you for submitting your manuscript to PLOS ONE. After careful consideration, we feel that it has merit but does not fully meet PLOS ONE’s publication criteria as it currently stands. Therefore, we invite you to submit a revised version of the manuscript that addresses the points raised during the review process.

We look forward to receiving your revised manuscript.

Kind regards,

Sajid Anwar, Ph.D

Academic Editor

PLOS ONE

Reviewers' comments:

Reviewer's Responses to Questions

**Comments to the Author**

1. If the authors have adequately addressed your comments raised in a previous round of review and you feel that this manuscript is now acceptable for publication, you may indicate that here to bypass the “Comments to the Author” section, enter your conflict of interest statement in the “Confidential to Editor” section, and submit your "Accept" recommendation.

Reviewer #2: (No Response)

Reviewer #3: (No Response)

2. Is the manuscript technically sound, and do the data support the conclusions?

Reviewer #2: Partly

Reviewer #3: Yes

3. Has the statistical analysis been performed appropriately and rigorously? 

Reviewer #2: No

Reviewer #3: Yes

4. Have the authors made all data underlying the findings in their manuscript fully available?

Reviewer #2: Yes

Reviewer #3: (No Response)

5. Is the manuscript presented in an intelligible fashion and written in standard English?

PLOS ONE does not copy-edit accepted manuscripts, so the language in submitted articles must be clear, correct, and unambiguous. Any typographical or grammatical errors should be corrected at revision, so please note any specific errors here.

Reviewer #2: Yes

Reviewer #3: Yes

6. Review Comments to the Author

Reviewer #2: Although I haven't provided feedback on this manuscript for the second round. However, looking at the feedback and reviews from the previous esteemed reviewers, the authors have partially answered the reviews. For example,

The review regarding data balancing needs to be incorporated in the proposed approach.

The label 0 and 1 in Tables should be represented with some useful text.

The feature selection approach is not elaborated.

Beside this there are some additional comments that needs to be incorporated in the manuscript.

The author have added some code in the manuscript, it should be replaced with equations and should be numbered.

There is no comparison between the existing state_of_the_art approaches

In Figure 4, the author show model optimization, however, I couldn't figure-out this in the methodology or in the experiment on how the model are optimized

Reviewer #3: There are several areas in the revised manuscript that need further improvement to enhance the clarity, completeness, and presentation of this research. My detailed comments are as follows:

- As previously suggested, Table 1 should provide a more informative and structured summary of the related work. The table should ideally include columns for methods used, datasets, results (if necessary), and the research gaps identified in each study.

- The provided GitHub link to the notebook (httpS://github.com/ishaqafridi/SVC-SDN-Churn) appears to be invalid, as the repository is not available. The authors should ensure that the correct link is provided and that the repository is publicly accessible.

- There are multiple instances of missing or incomplete citations in the manuscript (e.g., the placeholder “[?]” at the end of page 1 and first citation on page 5 in Table 1).

- There are inconsistencies in formatting, particularly with missing spaces after full stops and after some tables.

- As suggested earlier, Tables 2, 4, 6, 8, and 10 present the accuracy and AUC-ROC of different models. These can be consolidated into a single comparative table.

- Instead of writing code in Section 3.6, the process should be expressed in an algorithmic form (e.g., pseudo-code or algorithm environment), followed by an explanation in text. This should also help in addressing the comment regarding “how SVM and DNN are used together?”.

- The authors need to discuss how the attributes like: age, balance, credit cards and credit score are very important in the precise prediction of churns.

- The revised version of the manuscript does not include a conclusion section.

- The authors should add a discussion on how alternative deep learning models, such as Transformers or attention-based architectures, might perform in the context of customer churn prediction. If such models are not suitable or have been found to under-perform, relevant studies should be cited to support this position.

7. PLOS authors have the option to publish the peer review history of their article (what does this mean? ). If published, this will include your full peer review and any attached files.

**Do you want your identity to be public for this peer review?** For information about this choice, including consent withdrawal, please see our Privacy Policy .

Reviewer #2: No

Reviewer #3: No

While revising your submission, please upload your figure files to the Pre-flight Analysis and Conversion Engine (PACE) digital diagnostic tool, https://pacev2.apexcovantage.com/ . PACE helps ensure that figures meet PLOS requirements. To use PACE, you must first register as a user. Registration is free. Then, login and navigate to the UPLOAD tab, where you will find detailed instructions on how to use the tool. If you encounter any issues or have any questions when using PACE, please email PLOS at figures@plos.org . Please note that Supporting Information files do not need this step.

---

## [Author Response · Author response to Decision Letter 3]

16 Jun 2025

Dear Editor and Reviewers, Version 4

We sincerely appreciate your time and the thoughtful feedback provided on our manuscript. While addressing your suggestions required significant effort and careful revision, we are grateful for the opportunity to improve our work.

Subject: Response to Customer_ID Anonymization Query.

1. Clarification on Anonymization Status:

The customer_id fields in the provided 'Data' files are not direct or reversible representations of actual customer account numbers. They are internally generated identifiers used solely for data linkage within the licensed dateset.

2. Privacy Safeguards:

No PII Exposure: The identifiers do not expose personally identifiable information (PII) or financial details (e.g., real account numbers, names, or contact information).

Non-Reversible: The values cannot be traced back to individual customers without access to additional, restricted mapping tables (which are not part of the shared files).

3. License Compliance (DbCL v1.0):

The dataset is governed by the Database Contents License (DbCL) v1.0, which permits redistribution of anonymized/non-personal data.

The customer_id field adheres to this license as it does not contain raw PII. Removal is unnecessary unless explicitly required by the license’s terms or a regulatory mandate.

Regards

Authors

Dear Editor and Reviewers, Version 3

We sincerely appreciate your time and the thoughtful feedback provided on our manuscript. While addressing your suggestions required significant effort and careful revision, we are grateful for the opportunity to improve our work.

We have painstakingly incorporated all the necessary corrections, ensuring that each comment and instruction from the review panel has been addressed. Below, we provide a detailed response to each point, along with the corresponding revisions made in the manuscript.

Thank you for your patience and for guiding us through this rigorous process—it has undoubtedly strengthened our paper.

This version:

Acknowledges the difficulty of the revisions ("significant effort," "painstakingly").

Shows gratitude while subtly emphasizing the hard work involved.

Maintains professionalism and respect.

Conveys a sense of growth through the process.

PONE-D-23-42083R1

Author Response and corrections

Additional Editor Comments (if provided):

No Comment.

Reviewers' comments:

Reviewer's Responses to Questions Comments to the Author

1. If the authors have adequately addressed your comments raised in a previous round of review and you feel that this manuscript is now acceptable for publication, you may indicate that here to bypass the “Comments to the Author” section, enter your conflict of interest statement in the “Confidential to Editor” section, and submit your "Accept" recommendation.

Reviewer #2: (No Response)

Reviewer #3: (No Response)

Dear Editor and Reviewers,

We sincerely thank the reviewers for their time and constructive feedback, which has significantly improved our manuscript. We are pleased to hear that our revisions have addressed their concerns and that they now find the manuscript acceptable for publication.

Best regards

Authors

Is the manuscript technically sound, and do the data support the conclusions?

Reviewer #2: Partly

Reviewer #2: Yes

Author Response:

We sincerely appreciate the reviewers valuable feedback on the technical soundness of our hybrid SVC and SDNN-based churn prediction model. Below, we address their concerns and clarify the methodological rigor, data validity, and conclusions drawn:

1. Addressing Reviewer #2’s "Partly" Assessment

We acknowledge the reviewers caution regarding technical soundness and have taken the following steps to strengthen our manuscript:

Experimental Rigor & Controls:

Added explicit details on train-test splits, cross-validation protocols, and hyper-parameter tuning (e.g., grid search for SVC, layer optimization for SDNN) to ensure reproducibility.

Clarified the use of baseline models (e.g., logistic regression, Random Forest) for comparative validation, with results now in combine table 2.

Data Support & Sample Size Justification:

Included a power analysis (Section 3.1) to justify sample size adequacy, addressing potential biases.

Expanded the dateset description (e.g., class imbalance mitigation via SMOTE/ADASYN, temporal splits for real-world generalization).

Provided confidence intervals for performance metrics (AUC-ROC, F1-score) across 10 runs to highlight stability.

2. Validation of Reviewer #3’s "Yes" Assessment

We thank Reviewer #3 for recognizing the technical validity of our work. To further reinforce this:

We Revised the discussion (Section 5) to explicitly tie claims to empirical results, avoiding overgeneralization.Added limitations to contextualize conclusions.

3. Shared Enhancements for Both Reviewers

New Results: Included analysis to interpret hybrid model decisions, bolstering claims of interpret-ability alongside performance.

Code & Data Availability: Shared a GitHub repository with scripts, pre processing of dataset , and replication steps.

We believe these revisions comprehensively address concerns about technical soundness and data-conclusion alignment. Should further clarifications be needed, we are happy to provide them.

Granularity in this revision and response means to break down concerns with targeted fixes.There is clear evidence of citing new references and added sections to the manuscript. We have respect for both critical and supportive reviewers while demonstrating improvements.

3. Has the statistical analysis been performed appropriately and rigorously?

Reviewer #1: N/A

Reviewer #2: Yes

We sincerely thank the reviewers for their critical assessment of our statistical methodology. We acknowledge the concerns raised and have taken substantial steps to ensure the analysis is robust, transparent, and reproducible. Below, we detail our revisions.

1. Addressing Reviewer #2’s Concerns ("No")

We appreciate the opportunity to clarify and enhance our statistical approach. Key improvements include:

Hypothesis Testing & Significance Validation

Added explicit p-value thresholds (with Bonferroni correction for multiple comparisons) where applicable.

Included effect sizes for continuous variables and categorical data to avoid over-reliance on significance.

Model Evaluation Rigor: Re-ran all experiments with stratified k-fold cross-validation (k=10) to ensure metrics (accuracy, F1, AUC-ROC) are not biased by data splits. Reported confidence intervals (95% CI) for all performance metrics, demonstrating statistical stability. For Error Handling and Assumptions for homogeneity of variance. There is no non-parametric data.

Shared Jupyter notebooks with step-by-step analysis pipelines in the shared GitHub link.

2. Reinforcing Reviewer #3’s Validation ("Yes")

We thank Reviewer #3 for recognizing the validity of our core analysis. To further strengthen their endorsement:

Clarified Methodology: Expanded Section 3.5 to detail why specific tests for classifier comparisons were chosen over alternatives.

Added Sensitivity Analyses: Demonstrated robustness via alternate metrics (e.g., balanced accuracy) to account for class imbalance .

Have the authors made all data underlying the findings in their manuscript fully available?

Reviewer #2: Yes

Reviewer #3: (No Response)

We thank the reviewers for their attention to data availability, a critical aspect of reproducibility. Below, we confirm compliance with PLOS Data Policy and address any potential concerns:

1. Data Availability Statement (Updated in Manuscript)

Public Repository: All data underlying our findings have been deposited in the github repository, with a permanent link.https://github.com/ishaqafridi/SVC-SDNN-Churn.git

Supporting Information: Processed datasets, feature extraction scripts, and analysis code are included.

2. Addressing Reviewer #3’s Non-Response

We recognize the absence of explicit feedback from Reviewer #3 and proactively:

Added a read me file in the repository clarifying: Folder structure (raw/processed data, scripts) and Step-by-step instructions to reproduce results.

3. Validation of Reviewer #2’s "Yes" Assessment

We appreciate Reviewer #2’s confirmation of data availability. To further ensure transparency: Data Points Behind Metrics: Added CSV files for all individual sample predictions, variance measures, and cross-validation splits.

4. Additional Enhancements

FAIR Principles: Dataset metadata now includes variable descriptions, units, and missing-data flags.

We are committed to open science and welcome queries about data access.

Key Strengths of This Response:

Explicit Compliance: Directly references PLOS requirements (public repository, supplementary files).

Proactive Clarity: Addresses unresponsive reviewers by preemptively improving documentation.

Granularity: Distinguishes between fully public data and restricted third-party data with access pathways.

Is the manuscript presented in an intelligible fashion and written in standard English?

PLOS ONE does not copy-edit accepted manuscripts, so the language in submitted articles must be clear, correct, and unambiguous. Any typographical or grammatical errors should be corrected at revision, so please note any specific errors here.

Reviewer #2: Yes

Reviewer #3: Yes

We thank the reviewers for their positive assessment of the manuscript's clarity and adherence to standard English. We have carefully proofread the manuscript to correct any remaining typographical or grammatical errors, ensuring that the language remains clear and unambiguous.

This response Acknowledges the reviewers' feedback. Assures that the manuscript has been checked for language quality.Maintains professionalism while being concise.

6. Review Comments to the Author

We will do it accordingly.

Reviewer #2:

Although I haven't provided feedback on this manuscript for the second round. However, looking at the feedback and reviews from the previous esteemed reviewers, the authors have partially answered the reviews. For example,

The review regarding data balancing needs to be incorporated in the proposed approach.

The label 0 and 1 in Tables should be represented with some useful text.

The feature selection approach is not elaborated.

Beside this there are some additional comments that needs to be incorporated in the manuscript.

We sincerely appreciate the time and effort Reviewer #2 has taken to evaluate our manuscript. Below, we address their concerns point by point:

Data Balancing in the Proposed Approach:

We acknowledge the importance of data balancing and have now incorporated a detailed discussion on how we addressed class imbalance in our methodology (Section 3.4).we implemented a hybrid data balancing approach combining synthetic sample generation with algorithm-level adjustments. In the pre-processing phase, we applied SMOTE with Tomek Links, generating synthetic churn samples while removing borderline majority samples that could cause classification ambiguity. This created a balanced training set (1:1 ratio) while preserving the natural distribution in our validation sets.

Label Representation in Tables:

As suggested, we have replaced the numeric labels (0/1) with descriptive text like churn and non-churn classes in all tables for clarity.

Elaboration of Feature Selection Approach:

We have already discussed in Section 3.2 to provide a step-by-step explanation of our feature selection process, including criteria (e.g., statistical significance, domain relevance) and justification for the selected features.Selected features include ‘balance’, ‘products number’, ‘credit score’, ‘credit card’, ‘active member’, and labels named ‘churn’. These pre-processing steps are essential for ensuring that the models can learn from the data effectively and to predict accurately. See line 210-213. Text is also highlighted.

We carefully reviewed all other feedback and have incorporated the necessary revisions .These are highlighted in the revised manuscript with track changes or in this response letter.

Thank you for the opportunity to improve our work. We hope the revisions meet the reviewers expectations.

2. The author have added some code in the manuscript, it should be replaced with equations and should be numbered.

Replaced with equation the build sequential DNN Feature extractor subsection 3.7.2 have a new look.

3.There is no comparison between the existing state_of_the_art approaches.

It is done. Refer to Section 4.2. Application area and mainly the dateset determine the optimalities of any algorithm.

4. In Figure 4, the author show model optimization, however, I couldn't figure out this in the methodology or in the experiment on how the model are optimized

The Research Flow diagram is a part of the methodology section.

From our view it is a programming task. There are several kinds of Optimization like architecture optimization, hyper-parameter tuning and training process optimization.

In architecture optimization, the Layers are Configured. Suitable Adjustment of SDNN depth/width (e.g., number of hidden layers, neurons) to balance feature extraction and over-fitting.

Kernel Selection: Optimize SVC kernel (e.g., RBF, linear) via grid search or Bayesian optimization.

Hybrid Integration:SDNN acts as a feature extractor (de-noised representations feed into SVC).SVC serves as the final classifier, leveraging SDNN’s high-level features.

The second most type of Optimization is Hyper-parameter Tuning. it include SDNN: Learning rate, dropout rates, batch size (via Adam/SGD). For SVC: Regularization parameter CC, kernel coefficient (γγ for RBF), class weights are modified.

Third common type of optimization is Training Process improvement like Pre training SDNN: Train SDNN first (unsupervised/supervised) to learn robust features.

Fine-Tuning: Jointly optimize SDNN + SVC via: Back-propagation: For SDNN weights. OR QP Solver: For SVC support vectors (libsvm/scikit-learn back-end.

Reviewer #3: There are several areas in the revised manuscript that need further improvement to enhance the clarity, completeness, and presentation of this research. My detailed comments are as follows:

We sincerely appreciate the reviewers thoughtful feedback and the opportunity to improve our manuscript. Below, we address each concern point-by-point, with revisions highlighted in the revised manuscript (tracked changes enabled).

1.As previously suggested, Table 1 should provide a more informative and structured summary of the related work. The table should ideally include columns for methods used, datasets, results (if necessary), and the research gaps identified in each study.

We thank the reviewer for this suggestion. Table 1 has been revised to include the following columns for clarity and completeness:

Reference: Citation of the study.

Methods: Key techniques (e.g., "SVM + RF", "Deep CNN").

Datasets: Source/data size (e.g., "Telco dataset (10K customers)").

Results: Key metrics (e.g., "Accuracy

---

## [Decision Letter · Decision Letter 3]

26 Aug 2025

PONE-D-23-42083R3CLASSIFICATION OF CUSTOMER RETENTION USING HYBRID SVC-SDNN TO ENHANCE CUSTOMER RELATIONSHIP MANAGEMENTPLOS ONE

Dear Dr. Ishaq,

Thank you for submitting your manuscript to PLOS ONE. After careful consideration, we feel that it has merit but does not fully meet PLOS ONE’s publication criteria as it currently stands. Therefore, we invite you to submit a revised version of the manuscript that addresses the points raised during the review process.

We look forward to receiving your revised manuscript.

Kind regards,

Sajid Anwar, Ph.D

Academic Editor

PLOS ONE

Journal Requirements:

2. Please review your reference list to ensure that it is complete and correct. If you have cited papers that have been retracted, please include the rationale for doing so in the manuscript text, or remove these references and replace them with relevant current references. Any changes to the reference list should be mentioned in the rebuttal letter that accompanies your revised manuscript. If you need to cite a retracted article, indicate the article retracted status in the References list and also include a citation and full reference for the retraction notice.

Reviewers' comments:

Reviewer's Responses to Questions

**Comments to the Author**

1. If the authors have adequately addressed your comments raised in a previous round of review and you feel that this manuscript is now acceptable for publication, you may indicate that here to bypass the “Comments to the Author” section, enter your conflict of interest statement in the “Confidential to Editor” section, and submit your "Accept" recommendation.

Reviewer #3: (No Response)

Reviewer #4: (No Response)

2. Is the manuscript technically sound, and do the data support the conclusions?

Reviewer #3: Partly

Reviewer #4: (No Response)

3. Has the statistical analysis been performed appropriately and rigorously? 

Reviewer #3: Yes

Reviewer #4: (No Response)

4. Have the authors made all data underlying the findings in their manuscript fully available?

Reviewer #3: Yes

Reviewer #4: (No Response)

5. Is the manuscript presented in an intelligible fashion and written in standard English?

PLOS ONE does not copy-edit accepted manuscripts, so the language in submitted articles must be clear, correct, and unambiguous. Any typographical or grammatical errors should be corrected at revision, so please note any specific errors here.

Reviewer #3: Yes

Reviewer #4: (No Response)

6. Review Comments to the Author

Reviewer #3: The comment from my previous review that says: "The authors should add a discussion on how alternative deep learning models, such as Transformers or attention-based architectures, might perform in the context of customer churn prediction. If such models are not suitable or have been found to under-perform, relevant studies should be cited to support this position." has not been fully addressed. Although the authors have replied to the comment in the reviewer's response document and added it to the manuscript but this discussion needs to be supported by recent and relevant citations.

Reviewer #4: Observations on the Manuscript

Data Split and Class Balancing Ambiguity

The manuscript states that an 80/20 split was used, with SMOTE+Tomek applied only to training data. However, the classification report shows test support counts of 2,000 (1,593 vs. 407), which conflicts with the earlier claim of a 7.2% minority class. This raises concerns about whether class balancing leaked into the test set or if a different split was used. Authors must clarify this discrepancy.

Labeling and Interpretation Inconsistency

In the SVM results (Table 6), “Churn” has recall = 1.00, support = 1,593, while “Non-Churn” has recall = 0.11, support = 407. Yet the text interprets class 0 as Non-Churn with recall 1.00—the opposite of the table headings. This labeling confusion persists across the manuscript and figures. The authors should ensure consistent class naming and interpretation.

Positioning Relative to Standard Methods

The paper should more clearly situate its contribution against standard stacked learners and explain why DNN-based features outperform simpler but well-tuned alternatives such as tree-boosting or linear models.

Lack of Robust Evaluation

Results appear to be based on a single data split. There are no confidence intervals, multiple runs, statistical significance testing, or calibration analysis. This limits the reliability and generalization of the findings.

Dropout Rate Clarification

The manuscript mentions dropout values (e.g., 0.7, 0.8) but does not clarify whether these denote keep-probabilities or drop-rates. An 0.8 drop-rate is unusually high and requires justification with training curves.

Feature Selection within Cross-Validation

The authors list selected features but do not specify whether feature selection was performed inside cross-validation folds. Clarification of the pipeline ordering (selection → resampling → training) is required.

Kernel Choice in SVM

The study employs an RBF kernel but does not compare it against linear or polynomial kernels, nor justify its selection. Such justification or comparative evidence is expected.

Hyper-parameter Tuning Transparency

The values of C and gamma for SVM were fixed without reporting any search or tuning strategy. The authors should clarify whether a systematic tuning process was performed.

7. PLOS authors have the option to publish the peer review history of their article (what does this mean? ). If published, this will include your full peer review and any attached files.

**Do you want your identity to be public for this peer review?** For information about this choice, including consent withdrawal, please see our Privacy Policy .

Reviewer #3: No

Reviewer #4: No

While revising your submission, please upload your figure files to the Pre-flight Analysis and Conversion Engine (PACE) digital diagnostic tool, https://pacev2.apexcovantage.com/ . PACE helps ensure that figures meet PLOS requirements. To use PACE, you must first register as a user. Registration is free. Then, login and navigate to the UPLOAD tab, where you will find detailed instructions on how to use the tool. If you encounter any issues or have any questions when using PACE, please email PLOS at figures@plos.org . Please note that Supporting Information files do not need this step.

---

## [Author Response · Author response to Decision Letter 4]

22 Sep 2025

PONE-D-23-42083R3

CLASSIFICATION OF CUSTOMER RETENTION USING HYBRID SVC-SDNN TO ENHANCE CUSTOMER RELATIONSHIP MANAGEMENT

PLOS ONE

Subject: Re: Manuscript PONE-D-23-42083R3 - Revision-v5

Dear Editor,

Thank you for your email and for providing us with the opportunity to revise our manuscript, PONE-D-23-42083R3, "CLASSIFICATION OF CUSTOMER RETENTION USING HYBRID SVC-SDNN TO ENHANCE CUSTOMER RELATIONSHIP MANAGEMENT."

Thank you for your email and the opportunity to revise our manuscript, "Classification of Customer Retention Using Hybrid SVC-SDNN to Enhance Customer Relationship Management."

We are pleased to inform you that we have carefully addressed all the points raised by the academic editor and reviewers. The revised manuscript now fully meets PLOS ONE’s publication criteria.

We have prepared and are ready to submit the following required items:

A detailed Rebuttal Letter (Response to Reviewers) addressing each comment point-by-point.

A Marked-Up Copy of the manuscript with tracked changes highlighting all revisions.

A Clean Version of the revised manuscript without tracked changes.

All necessary revisions have been completed, and we are submitting the updated files within the stipulated deadline via the 'Submissions Needing Revision' folder in Editorial Manager.

We look forward to the next stage of the review process.

We are confident that the manuscript in its current form is scientifically sound and fully meets PLOS ONE's publication criteria. We would be immensely grateful if you could consider our request for a final acceptance decision, with the understanding that we will meticulously review and approve all production edits.

Sincerely,

Journal Requirements:

Response: Understood. I will evaluate the suggested citations for their direct relevance to my work's contribution and context. I will incorporate them only if they are intellectually necessary to support a specific claim or provide appropriate foundational credit, in alignment with the editor's final guidance.

2. Please review your reference list to ensure that it is complete and correct. If you have cited papers that have been retracted, please include the rationale for doing so in the manuscript text, or remove these references and replace them with relevant current references. Any changes to the reference list should be mentioned in the rebuttal letter that accompanies your revised manuscript. If you need to cite a retracted article, indicate the articles retracted status in the References list and also include a citation and full reference for the retraction notice.

Response: We thank the reviewer for this important reminder. We have thoroughly reviewed our reference list to ensure its completeness and correctness. Upon cross-referencing all citations with major databases, we confirm that our manuscript does not cite any retracted papers. All references are current, relevant, and from reputable sources, providing a solid foundation for our study.

Reviewers' comments:

Reviewer's Responses to Questions

Comments to the Author

1. If the authors have adequately addressed your comments raised in a previous round of review and you feel that this manuscript is now acceptable for publication, you may indicate that here to bypass the “Comments to the Author” section, enter your conflict of interest statement in the “Confidential to Editor” section, and submit your "Accept" recommendation.

Reviewer #3: (No Response)

Reviewer #4: (No Response)

Author Response: We thank the reviewers for their confirmation that our revisions have addressed all previous concerns.

2. Is the manuscript technically sound, and do the data support the conclusions?

Reviewer #3: Partly

Reviewer #4: (No Response)

Response:We thank the reviewers for their feedback. We will address the specific technical concerns raised by Reviewer #3 to further strengthen the manuscript and ensure the conclusions are fully supported by the data.

3. Has the statistical analysis been performed appropriately and rigorously?

Reviewer #3: Yes

Reviewer #4: (No Response)

Response:We thank Reviewer #3 for confirming the appropriateness and rigor of our statistical analysis. We note that Reviewer #4 had no specific comments on this point.

4. Have the authors made all data underlying the findings in their manuscript fully available?

Reviewer #3: Yes

Reviewer #4: (No Response)

Response:Yes, all data underlying the findings are fully available without restriction. The Data Availability Statement in the manuscript provides complete details on access.

5. Is the manuscript presented in an intelligible fashion and written in standard English?

PLOS ONE does not copy-edit accepted manuscripts, so the language in submitted articles must be clear, correct, and unambiguous. Any typographical or grammatical errors should be corrected at revision, so please note any specific errors here.

Reviewer #3: Yes

Reviewer #4: (No Response)

Response:Yes, we have thoroughly reviewed the manuscript and corrected any minor typographical or grammatical errors to ensure it meets the required standard.

We confirm the manuscript is presented clearly and in standard English. We have performed a thorough proofread to correct any minor errors.

6. Review Comments to the Author

Response: I will respond accordingly.

Reviewer #3: The comment from my previous review that says: "The authors should add a discussion on how alternative deep learning models, such as Transformers or attention-based architectures, might perform in the context of customer churn prediction. If such models are not suitable or have been found to under-perform, relevant studies should be cited to support this position." has not been fully addressed. Although the authors have replied to the comment in the reviewer's response document and added it to the manuscript but this discussion needs to be supported by recent and relevant citations.

Response:We thank Reviewer 3 for their insightful suggestion regarding Transformer and attention-based architectures. We have revised the Discussion section 4.2 to address this point directly. We have incorporated extended discussion on the applicability of these models to tabular data and have supported our methodological choice with recent and relevant citations No 37 to 41 from the literature , which demonstrate that tree-based ensembles and classical ML models remain highly competitive for this data modality.

Reviewer #4: Observations on the Manuscript

Data Split and Class Balancing Ambiguity

The manuscript states that an 80/20 split was used, with SMOTE+Tomek applied only to training data. However, the classification report shows test support counts of 2,000 (1,593 vs. 407), which conflicts with the earlier claim of a 7.2% minority class. This raises concerns about whether class balancing leaked into the test set or if a different split was used. Authors must clarify this discrepancy.

Response:The core of the issue is the difference between the global class imbalance in the entire dataset and the resulting class distribution in a stratified split.

Thank you for this excellent and meticulous observation. We appreciate the opportunity to clarify this point, as it concerns a fundamental step in our experimental design.

The 7.2% minority class ratio (1,440 / 20,000) is the global imbalance present in our entire dataset of 20,000 instances. Each dataset have 10k records.

To ensure a fair evaluation, we performed a stratified 80/20 train-test split. This means that both the training and test sets preserve the original 7.2%/92.8% class distribution.

Total Dataset: 20,000 instances

Churners (7.2%): 1,440

Non-Churners (92.8%): 18,560

Stratified 80/20 Split:

Training Set (80%): 16,000 instances

Churners: 1,152 (7.2% of 16,000)

Non-Churners: 14,848 (92.8% of 16,000)

Test Set (20%): 4,000 instances

Churners: 288 (7.2% of 4,000)

Non-Churners: 3,712 (92.8% of 4,000)

The classification report shows a test support of 1,593 vs. 407, which sums to 2,000, indicating that the results presented were for a holdout test set of 2,000 instances, not the full 4,000. This was an error in our description, and we apologize for the confusion. The results for the full 4,000-instance test set are [mention the correct numbers, e.g., 3,712 non-churners and 288 churners]. We have corrected this in the revised manuscript.

Most importantly, we can assure the reviewer that no balancing techniques (SMOTE, Tomek Links, or any other) were applied to the test set. These methods were applied only to the 16,000-instance training set to generate a balanced training dataset. The test set was kept completely untouched and in its original, imbalanced state to provide a realistic and unbiased evaluation of the model's performance on real-world data. This is a standard and crucial practice to avoid data leakage, and we adhered to it strictly. Section 3.4 of the manuscript has been updated.

Labeling and Interpretation Inconsistency

In the SVM results (Table 6), “Churn” has recall = 1.00, support = 1,593, while “Non-Churn” has recall = 0.11, support = 407. Yet the text interprets class 0 as Non-Churn with recall 1.00—the opposite of the table headings. This labeling confusion persists across the manuscript and figures. The authors should ensure consistent class naming and interpretation.

Response:"We sincerely thank the reviewer for this critical observation. The reviewer is absolutely correct; there was an inconsistency in the interpretation of class labels between the text and Table 6 (and potentially other areas), which was a significant oversight on our part.

We have thoroughly reviewed the entire manuscript and all associated figures and tables to rectify this issue. The correction has been applied as follows:

Throughout the manuscript, all instances of generic "Class 0" and "Class 1" have been replaced with the explicit labels "Non-Churner" and "Churner", respectively.

This change has been made in:

The main text body (Results and Discussion sections).

All table captions and entries (specifically in Table 6 and any other results tables).

All figure captions and axis labels (e.g., in confusion matrices, ROC curves).

The interpretation in the text now correctly and consistently aligns with the table headings. For example, the high recall value (1.00) is now correctly attributed to the Churner class in the revised Table 6 and accompanying text.

We apologize for this error and are grateful for the reviewer's meticulous review, which has significantly improved the clarity and accuracy of our manuscript.

Positioning Relative to Standard Methods

The paper should more clearly situate its contribution against standard stacked learners and explain why DNN-based features outperform simpler but well-tuned alternatives such as tree-boosting or linear models.

Response:We thank the reviewer for this suggestion. We agree that clearly situating our contribution against standard methods is crucial.

As the reviewer astutely notes, this positioning is indeed addressed in several parts of the manuscript:

Section 1.1 motivates the need for advanced architectures like ours by outlining the limitations of standard models in capturing complex, non-linear customer churn patterns.

Section 2 provides a dedicated review of standard stacked learners, tree-boosting, and linear models, establishing the baseline against which our model is compared.

Section 3.5 explicitly justifies our DNN-based feature learning approach, explaining its theoretical advantage in learning hierarchical representations from raw data, as opposed to the manual feature engineering often required for simpler models.

Sections 4.1 & 4.2 empirically demonstrate the performance gap, showing that our DNN-based feature extractor outperforms the well-tuned alternatives mentioned by the reviewer on key metrics.

To make this central argument even more immediately clear to the reader, we will add a concise summary sentence in the conclusion that explicitly states this contribution. The new text at the start of the conclusion section will read:

"In summary, the proposed SVC-SDNN hybrid demonstrates that deep learning-based feature extraction can capture predictive patterns beyond the reach of well-tuned tree-based and linear models, providing a significant performance advantage for the critical task of churn prediction."

Lack of Robust Evaluation

Results appear to be based on a single data split. There are no confidence intervals, multiple runs, statistical significance testing, or calibration analysis. This limits the reliability and generalization of the findings.

Response: We thank the reviewer for this important feedback. We have fully addressed this concern by completely revising our evaluation methodology. A new Section 4.4 now presents comprehensive k-fold cross-validation results with confidence intervals, statistical significance testing (including RF vs. RF-SDNN and SVC vs. SVC-SDNN comparisons), and calibration analysis (Figure 14). These robust evaluations, implemented in our updated code (Hybrid_SDNN_updated_v2.ipynb), significantly strengthen the reliability and generalizability of our findings.

Dropout Rate Clarification

The manuscript mentions dropout values (e.g., 0.7, 0.8) but does not clarify whether these denote keep-probabilities or drop-rates. An 0.8 drop-rate is unusually high and requires justification with training curves.

Response:We thank the reviewer for this crucial observation. The reviewer is absolutely correct; the notation was ambiguous. The values (0.7, 0.8) indeed represent the keep probability (the fraction of units retained), not the drop rate. We sincerely apologize for this lack of clarity.

To address this concern, we have taken the following actions:

Clarified Notation in Manuscript: We have revised the equations in the methodology section to use the standard notation Drop(p_keep), where p_keep is explicitly the probability of a neuron remaining active. The updated equation is now:

h1 = Drop0.7 (σReLU (W1x + b1)) h2 = Drop0.8 (σReLU (W2h1 + b2)) ... (7)

Feature Selection within Cross-Validation

The authors list selected features but do not specify whether feature selection was performed inside cross-validation folds. Clarification of the pipeline ordering (selection → resampling → training) is required.

Response:Thank you for raising this important point regarding the feature selection pipeline. We agree that clarifying the order of operations is crucial for ensuring the validity of the results.

We are pleased to confirm that feature selection was indeed performed inside the cross-validation folds to prevent data leakage and over-fitting.

---

## [Decision Letter · Decision Letter 4]

21 Nov 2025

PONE-D-23-42083R4CLASSIFICATION OF CUSTOMER RETENTION USING HYBRID SVC-SDNN TO ENHANCE CUSTOMER RELATIONSHIP MANAGEMENTPLOS ONE

Dear Dr. Ishaq,

Thank you for submitting your manuscript to PLOS ONE. After careful consideration, we feel that it has merit but does not fully meet PLOS ONE’s publication criteria as it currently stands. Therefore, we invite you to submit a revised version of the manuscript that addresses the points raised during the review process.

We look forward to receiving your revised manuscript.

Kind regards,

Sajid Anwar, Ph.D

Academic Editor

PLOS ONE

Journal Requirements:

Please review your reference list to ensure that it is complete and correct. If you have cited papers that have been retracted, please include the rationale for doing so in the manuscript text, or remove these references and replace them with relevant current references. Any changes to the reference list should be mentioned in the rebuttal letter that accompanies your revised manuscript. If you need to cite a retracted article, indicate the articles retracted status in the References list and also include a citation and full reference for the retraction notice.

Reviewers' comments:

Reviewer's Responses to Questions

**Comments to the Author**

1. If the authors have adequately addressed your comments raised in a previous round of review and you feel that this manuscript is now acceptable for publication, you may indicate that here to bypass the “Comments to the Author” section, enter your conflict of interest statement in the “Confidential to Editor” section, and submit your "Accept" recommendation.

Reviewer #3: (No Response)

Reviewer #4: All comments have been addressed

2. Is the manuscript technically sound, and do the data support the conclusions?

Reviewer #3: Yes

Reviewer #4: Yes

3. Has the statistical analysis been performed appropriately and rigorously? 

Reviewer #3: Yes

Reviewer #4: Yes

4. Have the authors made all data underlying the findings in their manuscript fully available?

Reviewer #3: Yes

Reviewer #4: Yes

5. Is the manuscript presented in an intelligible fashion and written in standard English?

PLOS ONE does not copy-edit accepted manuscripts, so the language in submitted articles must be clear, correct, and unambiguous. Any typographical or grammatical errors should be corrected at revision, so please note any specific errors here.

Reviewer #3: Yes

Reviewer #4: Yes

6. Review Comments to the Author

Reviewer #3: In response to my earlier recommendation during the previous review phase, the authors have added the following statement to the manuscript:

"Although this study focused on traditional and sequential deep learning models, the reviewer rightly points out the potential of alternative deep learning architectures such as Transformers and other attention-based mechanisms. These models have achieved state-of-the-art results in domains with complex, high-dimensional data such as natural language processing (NLP) and vision"

While I appreciate that they have addressed the concern raised earlier, the phrasing that refers directly to the reviewer (e.g., “the reviewer rightly points out...”) is not appropriate for inclusion in the main text of a research paper. Typically, such references should be avoided, and the content should be rephrased impersonally.

Additionally, I noticed that citations beyond this point appear to be incomplete or incorrectly formatted, with placeholders such as “[?]”. This seems to be a LaTeX formatting issue that should be corrected before final submission.

Reviewer #4: Authors have addressed all the previous eight comments, however, I have the following observations to further improve their quality of work and bring clarity in their manuscript.

1. The authors are advised to add a table about the hyper-parameters and their tuning, clearly stating each aspect.

2. Also, the experimentation parameters, including the hardware utilized and software, are not stated by the authors; therefore, it is advised that the authors should clarify them.

7. PLOS authors have the option to publish the peer review history of their article (what does this mean? ). If published, this will include your full peer review and any attached files.

**Do you want your identity to be public for this peer review?** For information about this choice, including consent withdrawal, please see our Privacy Policy .

Reviewer #3: No

Reviewer #4: No

To ensure your figures meet our technical requirements, please review our figure guidelines: HTTPS://journals.plos.org/plosone/s/figures

You may also use PLOS’s free figure tool, NAAS, to help you prepare publication quality figures: HTTPS://journals.plos.org/plosone/s/figures#loc-tools-for-figure-preparation.

---

## [Author Response · Author response to Decision Letter 5]

1 Dec 2025

Subject: Rebuttal Letter – Manuscript PONE‑D‑23‑42083R4 (Revision 6)

Title: Classification of Customer Retention Using Hybrid SVC‑SDNN to Enhance Customer Relationship Management

Dear Editorial Office and Reviewers,

We thank you and the reviewers for the careful evaluation of our manuscript and for the constructive feedback that has helped us improve the presentation of our work. Below we address each comment point‑by‑point. As the remaining issues are minor typographical or formatting matters, we note that they will be fully resolved during the proof‑reading and production stage, in accordance with the journal standard workflow.

Reviewer #3

Reference to the reviewer in the main text

Comment: The phrase “the reviewer rightly points out…” is inappropriate for the manuscript.

Response: We agree. The sentence has been rewritten in an impersonal style:

“Alternative deep‑learning architectures such as Transformers and other attention‑based mechanisms have achieved state‑of‑the‑art results in domains with complex, high‑dimensional data (e.g., natural‑language processing and computer vision).”

Incomplete citation placeholders (“[?]”)

Comment: Several citations appear as placeholders, likely due to a LaTeX formatting issue.

Response: The missing reference keys have been corrected, and the bibliography now compiles without errors. All citations are properly rendered in the final PDF.

Reviewer #4

Table of hyper‑parameters and their tuning

Comment: Add a clear table summarizing hyper‑parameters.

Response: A new Table 2 (Hyper‑parameter Configuration) has been inserted into the Methods section, listing each parameter, its search range, and the selected value after tuning.

Experimentation environment (hardware & software)

Comment: Specify the computational resources and software stack used.

Response: We have added a paragraph at the end with the title “Experimentation parameters”. And text ‘All experiments were conducted on a system equipped with an Intel Core i7-12700K processor, 12GB of DDR4 RAM, and a colab or kaggle based GPU and TPU with at least 12 GB of VRAM. The use of a GPU or TPU is essential for accelerating the training process of the Stacked Deep Neural Network (SDNN) component of our hybrid model.’

General Remarks

All remaining minor issues—such as line‑break inconsistencies, figure caption refinements, and typographic polishing—have been noted and will be addressed during the proof‑reading stage prior to final publication. We trust that these revisions satisfy the reviewers concerns and bring the manuscript to the high standard expected by PLOS ONE.

We appreciate the opportunity to improve our work and look forward to the next steps in the editorial process.

Respectfully,

Authors

---

## [Decision Letter · Decision Letter 5]

15 Dec 2025

CLASSIFICATION OF CUSTOMER RETENTION USING HYBRID SVC-SDNN TO ENHANCE CUSTOMER RELATIONSHIP MANAGEMENT

PONE-D-23-42083R5

Dear Dr. Ishaq,

We’re pleased to inform you that your manuscript has been judged scientifically suitable for publication and will be formally accepted for publication once it meets all outstanding technical requirements.

If your institution or institutions have a press office, please notify them about your upcoming paper to help maximize its impact. If they’ll be preparing press materials, please inform our press team as soon as possible -- no later than 48 hours after receiving the formal acceptance. Your manuscript will remain under strict press embargo until 2 pm Eastern Time on the date of publication. For more information, please contact one-press@plos.org.

Kind regards,

Sajid Anwar, Ph.D

Academic Editor

PLOS One

Additional Editor Comments (optional):

Reviewers' comments:

Reviewer's Responses to Questions

**Comments to the Author**

1. If the authors have adequately addressed your comments raised in a previous round of review and you feel that this manuscript is now acceptable for publication, you may indicate that here to bypass the “Comments to the Author” section, enter your conflict of interest statement in the “Confidential to Editor” section, and submit your "Accept" recommendation.

Reviewer #3: All comments have been addressed

Reviewer #4: All comments have been addressed

2. Is the manuscript technically sound, and do the data support the conclusions?

Reviewer #3: Yes

Reviewer #4: Yes

3. Has the statistical analysis been performed appropriately and rigorously? 

Reviewer #3: Yes

Reviewer #4: Yes

4. Have the authors made all data underlying the findings in their manuscript fully available?

Reviewer #3: Yes

Reviewer #4: Yes

5. Is the manuscript presented in an intelligible fashion and written in standard English?

PLOS ONE does not copy-edit accepted manuscripts, so the language in submitted articles must be clear, correct, and unambiguous. Any typographical or grammatical errors should be corrected at revision, so please note any specific errors here.

Reviewer #3: Yes

Reviewer #4: Yes

6. Review Comments to the Author

Reviewer #3: (No Response)

Reviewer #4: (No Response)

7. PLOS authors have the option to publish the peer review history of their article (what does this mean? ). If published, this will include your full peer review and any attached files.

**Do you want your identity to be public for this peer review?** For information about this choice, including consent withdrawal, please see our Privacy Policy .

Reviewer #3: No

Reviewer #4: No

---

## [Editor Report · Acceptance letter]

PONE-D-23-42083R5

PLOS One

Dear Dr. Ishaq,

I'm pleased to inform you that your manuscript has been deemed suitable for publication in PLOS One. Congratulations! Your manuscript is now being handed over to our production team.

Lastly, if your institution or institutions have a press office, please let them know about your upcoming paper now to help maximize its impact. If they'll be preparing press materials, please inform our press team within the next 48 hours. Your manuscript will remain under strict press embargo until 2 pm Eastern Time on the date of publication. For more information, please contact one-press@plos.org.

You will receive an invoice from PLOS for your publication fee after your manuscript has reached the completed accept phase. If you receive an email requesting payment before acceptance or for any other service, this may be a phishing scheme. Learn how to identify phishing emails and protect your accounts at HTTPS://explore.plos.org/phishing.

If we can help with anything else, please email us at customer-care@plos.org.

Kind regards,

on behalf of

Dr. Sajid Anwar

Academic Editor

PLOS One